# Breaking the curse of dimensionality in structured density estimation

**Robert A. Vandermeulen**[*]      **Wai Ming Tai**[†]      **Bryon Aragam**[‡]

## Abstract

We consider the problem of estimating a structured multivariate density, subject to Markov conditions implied by an undirected graph. In the worst case, without Markovian assumptions, this problem suffers from the curse of dimensionality. Our main result shows how the curse of dimensionality can be avoided or greatly alleviated under the Markov property, and applies to arbitrary graphs. While existing results along these lines focus on sparsity or manifold assumptions, we introduce a new graphical quantity called "graph resilience" and show how it controls the sample complexity. Surprisingly, although one might expect the sample complexity of this problem to scale with local graph parameters such as the degree, this turns out not to be the case. Through explicit examples, we compute uniform deviation bounds and illustrate how the curse of dimensionality in density estimation can thus be circumvented. Notable examples where the rate improves substantially include sequential, hierarchical, and spatial data.

## 1 Introduction

Density estimation is a classical problem in statistical machine learning, and provides the backbone of modern generative models such as diffusion models, which are now state-of-the-art density estimators for a variety of applications, as well as normalizing flows, energy-based models, and implicit generative models. At the same time, density estimation is a notoriously difficult problem in high-dimensions, known to suffer from the so-called curse of dimensionality. When the density depends on only a few variables or, more generally, is supported on a low-dimensional manifold, it is known that the curse of dimensionality can be circumvented by substituting the ambient dimension $d$ with the effective dimension $s$ (e.g. Lafferty and Wasserman, 2008; Yang and Tokdar, 2015). But what happens when the distribution is spread over the entire space in a structured manner—is it still possible to circumvent the curse of dimensionality?

Three representative examples are given in Figure 1, corresponding to sequential, hierarchical, and spatial (or convolutional) data. In these examples, although both manifold and sparsity assumptions are violated, there are structured dependencies that one might hope to leverage when estimating the underlying density. These kinds of structures are pervasive in machine learning applications. For one example, consider computer vision and imaging. Images have long been modeled as a grid graph where adjacent pixels correspond to adjacent vertices (see Keener (2010), for example). Such an assumption is very natural: pixels tend to be strongly dependent on nearby pixels and independent of far away pixels. See Figure 2 for an example of this.

These examples are particularly appealing in applications, however, we emphasize that our problem setting is considerably more general, and applies to general dependence structures given by any

---

[*]Much of this work was conducted at the Berlin Institute for the Foundations of Learning and Data (BIFOLD), Technische Universität Berlin. Contact: `robert.anton.vandermeulen@gmail.com`

[†]The work was done when the author was at Nanyang Technological University. Contact: `taiwaiming2003@gmail.com`

[‡]University of Chicago. Contact: `bryon@chicagobooth.edu`

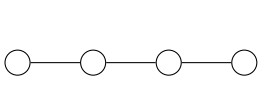 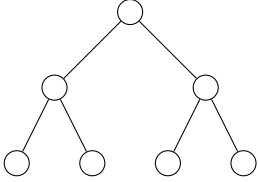 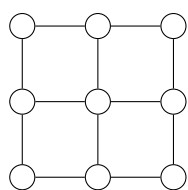

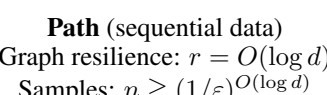

**Path** (sequential data)          **Tree** (hierarchical data)          **Grid** (spatial data)
Graph resilience: $r = O(\log d)$    Graph resilience: $r = O(1)$          Graph resilience: $r = o(d)$
Samples: $n \gtrsim (1/\varepsilon)^{O(\log d)}$    Samples: $n \gtrsim (1/\varepsilon)^{O(1)}$    Samples: $n \gtrsim (1/\varepsilon)^{o(d)}$

Figure 1: Examples of common structures that yield improvements in density estimation. As indicated by the path example on the left, which is also a tree, the worst-case resilience of any tree is $r = O(\log d)$, but for bounded-depth trees, $r = O(1)$.

Markov random field. Since real data is expected to (approximately) exhibit these types of structures, a reasonable question to ask is whether or not the curse of dimensionality persists in such structured settings.

**Structured density estimation**    To formalize the notion of structured dependencies in a high-dimensional, multivariate distribution, we adopt the framework of undirected graphical models, also known as Markov random fields (MRFs). In this setting, we are given samples from an unknown distribution $P$, with density $p$, over the random vector $X = (X_1, \ldots, X_d)$, and it is assumed that $P$ is Markov to some undirected graph $G$ (see Section 3.1 for definitions). We treat both cases where $G$ is *a priori* known, and where it is unknown but is in some known subset of all graphs. The graph $G$ encodes the underlying dependence structure between the variables, which we hope simplifies the estimation problem. For example, when $P$ is a Gaussian, this gives rise to the well-known Gaussian graphical model (Speed and Kiiveri, 1986), and various extensions of this idea to nonparametric settings are known, including trees (Liu et al., 2011; Györfi et al., 2022) and nonparanormal models (Liu et al., 2009). While this line of work also discusses the problem of *structure learning*, our focus is on the problem of nonparametric *density estimation*, which is comparatively understudied in graphical models. This may come as a surprise given the outsized literature on the general density estimation problem; see Section 2 for an overview of related work. In contrast to most existing work on density estimation, in lieu of imposing parametric or functional restrictions on $P$, the only assumption we make in addition to the Markov assumption is Lipschitz continuity.

**Overview**    Our main result establishes the sample complexity of estimating such a density—for arbitrary graphs $G$—in total variation (TV) distance: It is approximately $(1/\varepsilon)^{r+2}$, where $r \ll d$ is a novel graphical parameter we call *graph resilience* that depends only on $G$. Roughly, $r$ is a measure of how connected the graph $G$ is; the easier it is to disconnect $G$, the smaller $r$ will be. The examples in Figure 1 illustrate a range of values from constant $r = O(1)$ to sublinear $r = O(\sqrt{d}) = o(d)$. In the former case, this leads to an exponential improvement in the sample complexity over the usual nonparametric sample complexity of $(1/\varepsilon)^d$.

While it is not surprising that graphical structure (i.e. sparsity in the form of conditional independence) can make estimation easier, what is surprising is the quantity involved: It is not, as one might guess, one of the "usual" suspects such as sparsity, degree, or width. To capture the effective dimension of the problem and its resulting sample complexity, we introduce the concept of *graph resilience*. In particular, the usual suspects are insufficient to break the curse of dimensionality, whereas graph resilience does. Moreover, it is easy to construct examples where these are not only insufficient, but wholly misleading: Graph resilience can be controlled in graphs with unbounded degree, sparsity, and/or diameter.

Our work also marks a substantial departure from the existing literature that focuses primarily on functional restrictions (e.g. compositional structure), sparsity (e.g. density regression), or low-dimensional embeddings (e.g. manifold hypothesis). Instead, we impose no *explicit* restrictions on the functional form (although certain restrictions are implicit through the Markov assumption). In practice, empirical evidence points to a combination of these properties prevailing in real-world data,

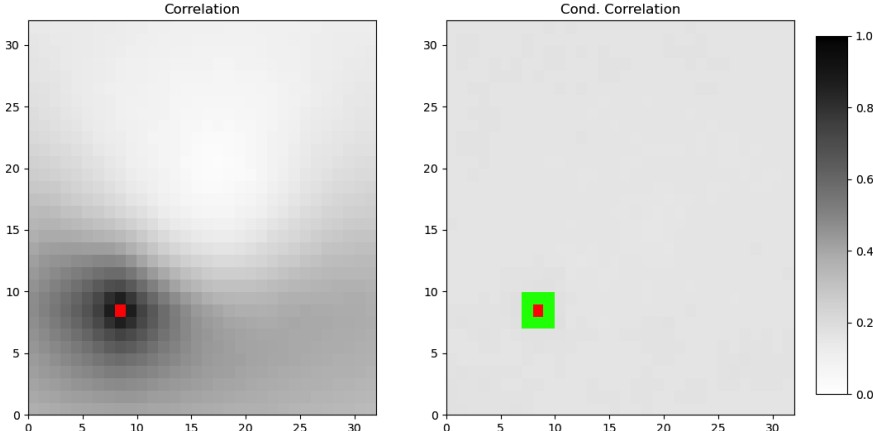

Figure 2: Heatmaps of the magnitude of the correlation between red pixel and every other pixel, using the CIFAR-10 training set. The left image shows the correlation without conditioning, the right image shows correlation conditioned on the green pixels. We see that the modeling the image as a Markov random grid is valid.

and thus our approach hopefully serves to provide another practical assumption under which density estimation is feasible, and in particular, the curse of dimensionality can be avoided.

**Contributions**   More precisely, we make the following contributions.

1. (Section 3.2) We introduce the graphical property of *resilience* (Definition 3.2), which approximately quantifies the connectivity of an undirected graph. Resilience is defined through the process of disintegration (Definition 3.1), which is described in detail.

2. (Section 3) We prove that the sample complexity of estimating a density $p$ that is Markov to any undirected graph $G$ scales with the resilience $r = r(G)$, as opposed to the dimension $d$ (Theorem 3.1). We also provide examples to show that other metrics such as degree, diameter, and sparsity cannot properly capture the sample complexity (Section 4.3). We also show that efficient estimation is still possible even if $G$ is unknown (Theorem 3.2).

3. (Section 4) We demonstrate numerous concrete examples where the resilience (and hence the sample complexity) can either be exactly calculated or bounded. These examples include familiar graphs such as trees (including paths for sequential data), cliques, and grids (also known as lattice graphs), and represent a broad continuum of possible complexities (Section 4.2).

All told, the potential savings implied by our results can be substantial, and are not isolated or pathological in any way: If there is a *single* independence relation satisfied by $P$, the effective dimension will be strictly less than the ambient dimension $d$, and in practical applications such as spatial or imaging data, there is an exponential savings in the sample complexity (see Figure 1). As we show, the graph resilience reveals a continuum of complexities ranging from dimension-independent (i.e. $r = O(1)$), in which case the curse of dimensionality is circumvented completely, to dimension-dependent with nontrivial savings (e.g. $r = o(d)$).

## 2   Related work

We begin by recalling classical rates and results on density estimation. The standard nonparametric rate for estimating an $L$-Lipschitz density $p$ in $d$ dimensions in TV distance is $n^{-1/(d+2)}$; see Devroye and Gyorfi (1985); Tsybakov (2009); Giné and Nickl (2015) for more details. This rate ignores dimension-dependent constants that affect finite-sample rates, and a more refined bound on the sample

complexity (ignoring log-factors) is given in McDonald (2017):

$$n \gtrsim \frac{d^d}{\varepsilon^{d+2}}. \tag{1}$$

See also Ghorbani et al. (2020); Jiao et al. (2023). For comparison, our main result is that only

$$n \gtrsim \frac{r d^{r/2+1}}{\varepsilon^{r+2}} \tag{2}$$

samples are needed (again up to log-factors) when $p$ is Markov to an undirected graph $G$ with resilience $r = r(G)$, and this rate cannot be improved among graphs whose resilience is at most $r$. The improvement over (1) is clear: Not only the exponent, which dominates the rate, but also the constant is improved by replacing $d$ with $r$, which can be much smaller than $d$ (see examples in Section 4.2).

Ignoring the dimension-dependent constant factor (as most papers do), the basic idea behind most results on circumventing the curse of dimensionality is to replace the $d$-dependence in the exponent of (1) with some $s < d$, where $s$ is the *effective dimension* of the problem. Examples include sparsity, low-dimensional embeddings (e.g. manifold assumptions), and hierarchical and/or compositional structure. Viewed from this perspective, our main contribution is to propose a new measure of effective dimension in structured data, where $s = r$ is the graph resilience.

Recently, there has been a renewed interest in this problem along two broad axes: 1) Generative models as density estimators, and 2) Breaking the curse of dimensionality. In the remainder of this section, we review this related work.

## 2.1 Density estimation

As noted in the introduction, density estimation is a classical problem with a literature dating back more than 50 years. Notably, Stone (1980, 1982) established minimax rates for nonparametric estimation problems including density estimation and regression. These papers derived the now-classical nonparametric rate $n^{-\beta/(2\beta+d)}$ for $\beta$-smooth densities, which implies the curse of dimensionality, i.e. unless $p$ is very smooth, then the sample complexity of estimating $p$ is exponential in the dimension. Yet, at the same time, the stark practical success of generative models suggest that high-dimensional density estimation may not be quite as intractable as this slow rate suggests. Motivated by these empirical observations, a growing line of work has established minimax optimality for a range of generative models, including GANs (Liang, 2017; Singh et al., 2018; Singh and Póczos, 2018; Uppal et al., 2019, 2020), diffusion models (Oko et al., 2023; Zhang et al., 2024; Cole and Lu, 2024; Tang and Yang, 2024), and variational autoencoders (Tang and Yang, 2021; Kwon and Chae, 2024).

Other theoretical developments have focused on efficient algorithms (Acharya et al., 2021, 2017; Chan et al., 2014) in the univariate case.

There has also been recent interest in developing density estimators that exploit graphical structure (Germain et al., 2015; Johnson et al., 2016; Khemakhem et al., 2021; Wehenkel and Louppe, 2021; Chen et al., 2024). Of course, there is an enormous literature on algorithms and methods for general density estimation that we cannot cover here.

Most closely related to our work are the papers Liu et al. (2007, 2011); Györfi et al. (2022). Liu et al. (2007) use the RODEO estimator on a model that satisfies a sparsity assumption, i.e. only $s \ll d$ variables are involved in the nonparametric component. Liu et al. (2011) use forests to approximate the underlying density under certain regularity conditions; Györfi et al. (2022) relax these conditions and replace forests with trees. Their main result is a pointwise $O(n^{-1/4})$ rate of convergence for estimating a tree-structured density, which is notably dimension-independent. The main difference between our results and these previous results is that our results apply to general graphs $G$ that may not be trees or forests, in addition to being *uniform* in $p$ and unimprovable (cf. Remark 1 in Györfi et al., 2022, in particular). Most importantly, moving beyond tree-based models requires new ideas, and motivates our introduction of the graph resilience to measure the effective dimension of the estimation problem.

After the initial posting of our paper, we were made aware of the related work by Bos and Schmidt-Hieber (2023) which proposes a supervised approach to density estimation that also leverages the Markov assumption to break the curse of dimensionality in density estimation.

## 2.2 Curse of dimensionality

There is a long line of literature on understanding how and when the curse of dimensionality can be avoided. Common assumptions include the manifold assumption (Pelletier, 2005; Ozakin and Gray, 2009; Jiang, 2017; Schmidt-Hieber, 2019; Nakada and Imaizumi, 2020; Berenfeld et al., 2022; Jiao et al., 2023), additive structure (Stone, 1985; Raskutti et al., 2012), compositional structure (Horowitz and Mammen, 2007; Juditsky et al., 2009; Kohler and Krzyżak, 2017; Schmidt-Hieber, 2017; Bauer and Kohler, 2019; Kohler and Langer, 2021; Shen et al., 2021), low-rank structure (Hall and Zhou, 2003; Hall et al., 2005; Song and Dai, 2013; Amiridi et al., 2022a,b; Vandermeulen and Ledent, 2021; Vandermeulen, 2023) and sparsity (Liu et al., 2007; Lafferty and Wasserman, 2008; Yang and Tokdar, 2015). Bach (2017) showed that neural networks are adaptive to many of these underlying structures.

This line of work is particularly relevant as it pertains to breaking the curse of dimensionality via structural assumptions on the unknown parameter. Notably, it seems that the advantages of (in)dependence via the Markov property have not been thoroughly investigated. Our work aims to fill this gap for a wide range of structured models that *do not fit into* any of the classes above. Indeed, it is easy to construct densities that are non-sparse (i.e. every variable is active), non-additive and non-compositional (we consider arbitrary continuous densities), and are not supported on any lower-dimensional manifold, but that are Markov to a given graph $G$.

# 3 Main Results

Before presenting the main results of this paper we must present some basic background.

## 3.1 Background Definitions and Notation

Throughout the paper, we use undirected graphs to model the dependencies in $P$. We adopt the usual terminology and conventions from graphical models: $G = (V, E)$ is an undirected graph with $V = X = [d]$ and $d = \dim(X)$. To avoid technical complications, we assume compact support with $X = (X_1, \ldots, X_d) \in [0, 1]^d$. Two disjoint subsets $A, B \subset V$ are said to be separated by $C$ if all paths connecting $A$ to $B$ intersect $C$; equivalently, the subgraph over $(A \cup B) - C$ is disconnected. The distribution $P$ is called Markov with respect to $G$ if

$$A \text{ is separated from } B \text{ by } C \implies A \perp\!\!\!\perp_P B \mid C, \tag{3}$$

where $\perp\!\!\!\perp_P$ denotes conditional independence in $P$. In other words, graph separation implies conditional independence, but not necessarily vice versa. See Lauritzen (1996) for a review of graphical modeling terminology.

We term a subgraph of $G$ to be a *component of $G$* (sometimes called a *connected component*) if it is a maximal connected subgraph of $G$. A *path* in $G$ is a sequence of vertices $(v_0, v_1, \ldots, v_k)$ such that $v_i - v_{i+1}$ in $G$ (i.e. $(i, i + 1) \in E$). A *simple* path is a path without repeated vertices. The *length* of a path is the number of edges in the path; e.g. the length of $(v_0, v_1, \ldots, v_k)$ is $k$. A *geodesic* path between two vertices is any path of shortest length, and the distance between two vertices is the length of any geodesic between them. For graphs $G, G'$ the notation $G \oplus G'$ denotes a disjoint union of graphs, i.e., the graph whose vertex set is the disjoint union of the vertices in $G$ and $G'$ and inherits edges from the edges in $G$ and $G'$. For a graph $G = (V, E)$, for $V' \subset V$, the graph $G \setminus V'$ denotes the graph with vertices $V \setminus V'$ and the edges $\{i, j\} \in E$ where $\{i, j\} \subseteq V \setminus V'$.

For any $d \in \mathbb{N}$ and $L \geq 0$, let $\mathcal{D}_d$ be the set of densities on $[0, 1]^d$ and $\mathcal{D}_{d,L} \subset \mathcal{D}_d$ be those densities which are $L$-Lipschitz continuous. For any graph $G$ with $d$ vertices, we define $\mathcal{D}(G) \subseteq \mathcal{D}_d$, such that, for $p \in \mathcal{D}(G)$ and $(X_1, \ldots, X_d) \sim p$, the entries of the random variable, $X_1, \ldots, X_d$, satisfy the global Markov property with respect to the graph $G$. Finally let $\mathcal{D}_L(G) \triangleq \mathcal{D}(G) \cap \mathcal{D}_{d,L}$. When estimating these densities we will sometimes use the term *sample complexity*. This refers to the number of samples necessary to estimate a density to within $\varepsilon$ error in total variation distance. For functions, $f$ on $[0, 1]^d$ we define $\|f\|_1 = \int |f(x)| dx$; this is the total variation distance.

## 3.2 Graph Resilience

The key concept in this work, which characterizes the difficulty of estimating densities in $\mathcal{D}(G)$, is what we term the *graph resilience of $G$*. Graph resilience is based on a process we call a *disintegration*.

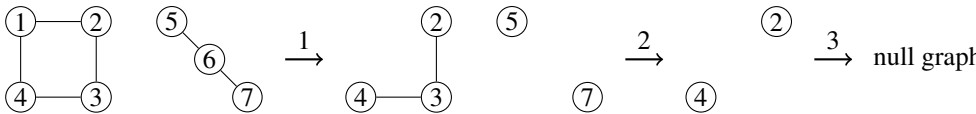

Figure 3: Visual representation of the steps of the 3-disintegration ($\{1,6\}, \{3,5,7\}, \{2,4\}$). In each step of the disintegration, one vertex is removed from every graph component. The total number of steps to the null graph is 3.

**Definition 3.1.** For a graph $G = (V, E)$, an $r$-tuple $(V_1, \ldots, V_r)$ with $V_i \subseteq V$ is called a *disintegration of $G$* if:

1. $\{V_i\}_i$ is a partition of $V$;

2. The elements of $V_1$ all lie in different components of $G$;

3. $|V_1|$ is equal to the number of components in $G$;

4. For all $i \in [r-1]$ the elements of $V_{i+1}$ lie in different components of $G \setminus \bigcup_{j=1}^{i} V_j$;

5. $|V_{i+1}|$ is equal to the number of components in $G \setminus \bigcup_{j=1}^{i} V_j$.

The *length* of a disintegration is the value $r$ in the above definition. A disintegration of length $r$ will be called an $r$-disintegration.

The first disintegration step, $V_1$ above, can be thought of as the process of picking one vertex from each component in $G$ and removing it. The next step of a disintegration, $V_2$ above, then selects and removes one vertex from each component of the resultant graph. An $r$-disintegration is a sequence of $r$ such steps until one is left with the null graph. It is possible for a graph to admit many different disintegrations of different lengths. A visual representation of the steps of a graph disintegration can be found in Figure 3.

The *resilience* of a graph $G$ describes the length of the shortest possible disintegration of $G$.

**Definition 3.2.** For a graph $G$, the *resilience of $G$*, denoted $r(G)$, is the smallest $r$ such that there exists a $r$-disintegration of $G$.

We elaborate more on some properties of graph resilience in Section 4, including upper bounds for the graph resilience of several graphs corresponding to image or sequence data. First, we need to establish that graph resilience is the right metric for quantifying the sample complexity of density estimation.

### 3.3 Sample complexity

The following theorem characterizes the difficulty of estimating a density which is known to satisfy the Markov property in terms of its graph. All proofs are deferred the appendix.

**Theorem 3.1.** *Let $G$ be a (known) graph whose number of vertices is $d$ and resilience is $r$. Let $L \geq 0$. For any $0 < \varepsilon < 1$, there exists an algorithm that takes $n = \Omega\left(\frac{r d^{r/2+1} L^r}{\varepsilon^{r+2}} \log(\frac{dL}{\varepsilon}) + \frac{\log(1/\delta)}{\varepsilon^2}\right)$ i.i.d. samples drawn from any $p \in \mathcal{D}_L(G)$ and returns a distribution $q$ such that*

$$\|p - q\|_1 \leq \varepsilon \quad \text{with probability at least } 1 - \frac{\delta}{3}.$$

This corresponds to a convergence rate of $\widetilde{O}(n^{-1/(r+2)})$ and is *uniform* over $\mathcal{D}_L(G)$. For comparison, the rate $O(n^{-1/(d+2)})$ is known to be optimal for estimating densities in $\mathcal{D}_{d,L}$ in the total variation norm (e.g. Beirlant and Gyorfi, 1998). Consequently the rate $\widetilde{O}(n^{-1/(r+2)})$ indicates that the rate of convergence when estimating densities in $\mathcal{D}_L(G)$ is akin to estimating densities in $\mathcal{D}_{r,L}$. We will see in Section 4.2 that, for graphs typically used to represent audio or image data, the effective dimension for estimating densities can be drastically, even exponentially, smaller than the ambient dimension.

While it may be reasonable to assume a known graph in some situations, one often encounters the situation where the graph is unknown. If one assumes that the graph $G$ is unknown, but that $G$ lies in some subset of graphs whose maximum graph resilience is bounded above by some value $r$ then it is still possible to achieve a rate of $\widetilde{O}(n^{-1/(r+2)})$.

**Theorem 3.2.** *Let $\mathcal{G}$ be the set of all graphs whose number of vertices is $d$ and resilience is $r$. Let $L \geq 0$. For any $0 < \varepsilon < 1$, there exists an algorithm that takes $n = \Omega\left(\frac{rd^{r/2+1}L^r}{\varepsilon^{r+2}}\log(\frac{dL}{\varepsilon}) + \frac{\log(1/\delta)}{\varepsilon^2}\right)$ i.i.d. samples drawn from any $p \in \bigcup_{G \in \mathcal{G}} \mathcal{D}_L(G)$ and returns a distribution $q$ such that*

$$\|p - q\|_1 \leq \varepsilon \quad \text{with probability at least } 1 - \frac{\delta}{3}.$$

We show in Appendix D that these rates are optimal (up to a polylogarithmic factor) for all dimensions $d$ and resiliences $r$. However, it's worth noting that these rates are not optimal for all graphs $G$. For instance, for the special case of trees see Györfi et al. (2022), where better rates can be achieved. The case of general graphs $G$ is an open problem.

### 3.4 Proof Outline and Practical Consequences

We will outline our proof techniques and demonstrate their relation to practically implementable estimators. Our theorem proofs employ disintegrations to construct a class of densities that closely approximate those with the given MRF. This class contains densities that take the form of histograms, i.e., piecewise constant densities on a grid. The remainder of the argument is relatively standard: A finite collection of the aforementioned class is found to cover the space, from which our estimate is chosen using a method akin to Scheffé tournaments (Scheffe, 1947; Yatracos, 1985; Devroye and Lugosi, 2001; Ashtiani et al., 2018). Disintegrations are the core novel aspect of this work, characterizing a class of histograms much smaller than the full set, thereby reducing its metric entropy and enhancing the sample efficiency of our selection.

At a high level, a disintegration outlines a method to estimate a density by inductively conditioning out the entries of a random vector. For instance, consider a random vector $[X_1, X_2]$. The disintegration in Figure 3.4 corresponds to first estimating a histogram for $X_2$, and then, for each bin $b$ of the $X_2$ histogram, estimating a histogram for

Figure 4: A simple disintegration example.

$X_1|X_2 \in b$. The resilience of a graph simply characterizes the shortest disintegration, i.e. the most efficient decomposition of a distribution into factors. Removing one vertex from each component of a graph captures the idea that, after sufficient conditioning, these components become independent. This allows us to avoid estimating the high-dimensional joint density of all vertices in the graph. Instead, we can estimate the low-dimensional components individually and take their product, which is effective due to the following inequality

$$\left\|\prod_{i=1}^{d} p_i - \prod_{j=1}^{d} q_j\right\|_1 \leq \sum_{i=1}^{d} \|p_i - q_i\|_1 \quad \text{for any probability densities } p_1, \ldots, p_d, q_1, \ldots, q_d.$$

Thus a disintegration can be thought of as a "meta-algorithm" providing an ordering for estimating conditional densities. In practice, it would be up to the implementation to determine how to handle the one-dimensional and conditional density estimation.

## 4 Graph Resilience Examples

Theorems 3.1 and 3.2 demonstrate that the graph resilience acts as the effective dimension of a nonparametric estimation problem, however, graph resilience is still somewhat of an opaque property. In this section we will go over basic properties of graph resilience and describe graph resiliences for some graphs that reflect real-world settings.

### 4.1 Graph Resilience Properties

Here we present some basic results regarding graph resilience. We first introduce two very basic lemmas outlining the most fundamental properties of graph resilience. The first lemma shows how graph resilience behaves with disjoint graph union.

| Graph $G$ | Resilience $r = r(G)$ | Sample complexity $n$ |
|---|---|---|
| Trees (depth $k$) | $\leq k$ | $\varepsilon^{-(k+2)}$ |
| Trees (general) | $O(\log(d))$ | $\sim \varepsilon^{-(\log(d)+2)}$ |
| Paths | $O(\log(d))$ | $\sim \varepsilon^{-(\log(d)+2)}$ |
| Grid | $O(\sqrt{d})$ | $\sim \varepsilon^{-(\sqrt{d}+2)}$ |
| Complete graph | $d$ | $\varepsilon^{-(d+2)}$ |

Table 1: Example graph resiliences and corresponding sample complexities

**Lemma 4.1.** *Let $G_1, \ldots, G_m$ be graphs, then $r(\bigoplus_{i=1}^{m} G_i) = \max_i r(G_i)$.*

This lemma encapsulates the notion that estimating joint density for a collection of independent random vectors, e.g., $(Y_1, \ldots, Y_m) \sim \prod_{i=1}^{m} \mu_i$, from a collection of samples is no more difficult than estimating each independent vector individually, $\widehat{\mu}_i \approx \mu_i$, and taking the product measure, $\prod_{i=1}^{m} \widehat{\mu}_i \approx \prod_{i=1}^{m} \mu_i$.

The second lemma demonstrates the behavior of graph resilience upon vertex removal.

**Lemma 4.2.** *Let $G = (V, E)$ be a graph and let $V' \subset V$, then $r(G) \leq r(G \setminus V') + |V'|$.*

This corresponds to conditioning the random variables in $V \setminus V'$ on the $|V'|$-dimensional random vector corresponding to $V'$.

Simpler graphs, in terms of number of edges and vertices, tend to have smaller graph resilience. The following lemma shows that adding edges to a graph will, at most, increase its resilience by the number of edges added.

**Lemma 4.3.** *Let $G = (V, E)$ be a graph, let $E'$ be a set of edges for vertices $V$, and let $G' = (V, E \cup E')$, then $r(G') \leq r(G) + |E'|$.*

For a pair of graphs $G, G'$, let the relation $G' \leq G$ denote that $G'$ is isomorphic to some subgraph of $G$. The following lemma demonstrates that removing random variables and dependencies between random variables can only reduce graph resilience.

**Lemma 4.4.** *For a pair of graphs $G$ and $G'$, if $G' \leq G$, then $r(G') \leq r(G)$.*

The following corollary follows directly from the previous Lemma 4.4 and Lemma 4.6, which we present later.

**Corollary 4.5.** *Let $G$ be a graph whose largest clique contains $k$ vertices, then $r(G) \geq k$.*

## 4.2 Example Graph Resiliences

From these properties of graph resilience we can derive the graph resilience of some example graphs. Some results from this section are summarized in Table 1. We begin with a couple of simple cases.

The following lemma demonstrates the surprising fact that, if even a single edge is missing from the graph, then the effective dimension of the estimation problem is strictly smaller than the ambient dimension $d$.

**Lemma 4.6.** *For a graph $G = (V, E)$, $r(G) = |V|$ if and only if $G$ is a complete graph.*

Analogously, a graph can only have resilience 1 if it is the empty graph.

**Lemma 4.7.** *For a graph $G = (V, E)$, $r(G) = 1$ if and only if $E = \emptyset$.*

The star graph is an example of a connected graph whose resilience is very small. Recall that a star graph $S_d$ is a graph with a single central vertex connected to every other node, i.e. $E = \{(i, j) : j \neq i\}$. Equivalently, it is (a) a complete bipartite graph with a single vertex in one partition or (b) a tree of depth one with a single root.

**Lemma 4.8.** $r(S_d) \leq 2$.

In the remainder of this section, we describe additional examples of graphs whose resilience can be bounded. We focus on four broad classes of graphs, categorized by the application of interest: Hierarchical data (e.g. language, biology, phylogeny), sequential data (e.g. audio, video, language), spatial data (e.g. images, vision, signal processing), and clustered data (e.g. genetics, ecology, medicine). See Figure 1 for a guide to these classes of graphs.

The following definition will be helpful for analyzing graphs that have a linear or grid shape.

**Definition 4.1.** For a graph $G$ and $n \in \mathbb{N}$, the power graph $G^n$, is the graph which inherits its vertices from $G$ and a pair of vertices in $G^n$ are adjacent when their distance in $G$ is at most $n$.

### 4.2.1 Hierarchical Data

Hierarchical data arises from distributions that have a tree-like structure, i.e. $G$ is tree. In the worst-case, the resilience of a tree can scale at most logarithmically with $d$, however, in practical applications where the tree has bounded depth the resilience is also bounded.

**Lemma 4.9.** *Let $G$ be a $k$-ary tree with depth $m$, then $r(G) \leq m$.*

**Lemma 4.10.** *Let $G$ be a tree with $d$ vertices, then $r(G) \leq \log_2(d) + 1$.*

### 4.2.2 Sequential Data

A path graph naturally represents sequential data. For a more realistic model one may assume that a vertex in a path graph is connected to all vertices within a given distance, so that nearby vertices are highly dependent, but far away indices are less dependent. Even with this assumption we see that, like tree graphs, the path graph's resilience only grows at rate $O(\log d)$.

**Definition 4.2.** The *path graph of length $d$*, denoted $L_d$, is the graph $(V, E)$ with $V = [d]$ and $E = \{\{i, j\} \mid |i - j| = 1\}$.

**Proposition 4.11.** *Recall that $L_d$ is the path graph of length $d$ as defined in Definition 4.2. For any $s, t \in \mathbb{N}$, we have*

$$r(L_{t(2^s-1)}^t) \leq st.$$

*Here, $L_{t(2^s-1)}^t$ is the power graph of $L_{t(2^s-1)}$ as defined in Definition 4.1.*

This, along with Lemma 4.4, yields the following rate on graph resilience for path graphs.

**Corollary 4.12.** *For any $d \in \mathbb{N}$ and any constant $t \in \mathbb{N}$, we have*

$$r(L_d^t) = O(t \log d).$$

### 4.2.3 Spatial Data

Image data is naturally represented via a grid graph. The following graph describes a $k \times k'$ grid of vertices with every vertex connected connected to its vertical, horizontal, and diagonal, neighbors.

**Definition 4.3.** The *grid graph of shape $k \times k'$*, denoted $L_{k \times k'}$, is the graph $(V, E)$ with $V = [k] \times [k']$ and $E = \{\{(i, j), (i', j')\} \mid (i, j) \neq (i', j'), |i - i'| \leq 1, |j - j'| \leq 1\}$.

**Proposition 4.13.** *Recall that $L_{k \times k}$ be the grid graph of shape $k \times k$ as defined in Definition 4.3. For any $s, t \in \mathbb{N}$, we have*

$$r(L_{t(2^s-1) \times t(2^s-1)}^t) \leq 4t^2 2^s.$$

*Here, $L_{t(2^s-1) \times t(2^s-1)}^t$ is the power graph of $L_{t(2^s-1) \times t(2^s-1)}$ as defined in Definition 4.1.*

Examples of a grid graph and its power can be found in Figure 5 in the appendix. The previous proposition gives us a general rate of $\sqrt{d}$ for grid graphs.

**Corollary 4.14.** *For any $k \in \mathbb{N}$ and any constant $t \in \mathbb{N}$, we have*

$$r(L_{k \times k}^t) = O(t^2 \sqrt{d}) \quad \text{where } d = k^2.$$

*Note that the graph $L_{k \times k}^t$ has $d$ vertices.*

#### 4.2.4 Clustered Data

Another common type of structured data exhibits clusters: Variables in the same cluster are dependent, while variables in different clusters are independent. This type of structure can be modeled with disjoint cliques, where each clique represents a cluster or community. Let $V_i \subset V$ denote each cluster, with $V_i \cap V_j = \emptyset$ and $V = V_1 \cup \cdots \cup V_m$ (i.e. $\{V_i\}$ is a partition of $V$). Let $d_i \triangleq |V_i|$, $G_i \triangleq K_{d_i}$ be a clique (complete graph) on $V_i$, and $G = \bigoplus_i G_i$. Then it follows from Lemmas 4.6 and 4.1 that $r(G) = \max_i d_i$. In particular, if $d_i = O(1)$, then $r(G) = O(1)$. Of course, this simply recovers the well-known result that a density that factorizes into a product of densities can be estimated at a rate that depends only on the largest factor. Using our graph resilience analysis we have that, for a stochastic block model, the graph resilience is bounded by the size of the blocks times the graph that encapsulates dependencies between the blocks.

**Lemma 4.15.** *Let $G' = ([k], E')$ be a graph. Let $G = (V, E)$ be a graph, such that there is a partition of $V = \bigcup_{i=1}^k V_i$, where, if $\{v, v'\} \in E$ it follows that $v \in V_i$ and $v' \in V_j$ where $i = j$ or $i$ and $j$ are adjacent in $G'$. Then $r(G) \leq r(G') \max_{i \in [k]} |V_i|$.*

This has the useful interpretation that the effect of additional dependencies *between* clusters does not interact with the clusters themselves. For example, if we allow for noisy interactions between clusters (e.g. as in a stochastic block model), the complexity scales separately with the noise and the size of the largest cluster.

### 4.3 Comparison to other graphical properties

The examples in the previous section can be used to show that the classical graphical properties that one might expect to govern the sample complexity surprisingly fail to capture the sample complexity.

**Degree** The first and most obvious is the maximum degree of $G$. The star graph $S_d$ is an example where the resilience is $O(1)$, and hence the sample complexity is $(1/\varepsilon)^{O(1)}$, but the maximum degree is $\Omega(d)$. Thus, the degree cannot properly capture the sample complexity. Moreover, the path graph shows the reverse: The maximum degree can be $O(1)$ while the resilience is $\Omega(\log d)$.

**Diameter** The diameter $\mathrm{diam}(G)$ of a graph $G$ is the length of its longest geodesic path. A clique $K_d$ thus has $\mathrm{diam}(K_d) = 1$ (since every node is connected to every other node), but $r(K_d) = d$. It is clear even from classical results that a clique cannot be estimated any faster than $(1/\varepsilon)^d$. Thus, the diameter also cannot capture the sample complexity.

**Sparsity** Call a density $p$ on $d$ variables $s$-sparse if $p(x) = q(x_S)$, where $S \subset [d]$ with $|S| = s$. Now suppose $p$ is $s$-sparse and $q$ is Markov to a star graph $S_s$ on $s$ vertices. Then Theorem 3.1 implies that $p$ can be estimated with $(1/\varepsilon)^{O(1)}$ samples (since $r(G) = O(1)$). Thus, taking $s \to \infty$, it follows that the sparsity $s$ cannot properly capture the sample complexity.

In other words, not only does the graph resilience properly capture the sample complexity of structured density estimation, it is also not simply a proxy for commonly used graphical properties.

## 5 Conclusion

This work has introduced a new concept, *graph resilience*, and demonstrated that it controls the complexity of estimating densities satisfying the Markov property. This characterization has shown that estimating such densities can be significantly easier than previous works have indicated. This finding contributes to the broader understanding of graph theoretical properties in statistical estimation and, more generally, to nonparametric estimation. Although our approach sheds light on the intrinsic possibilities and barriers to breaking the curse of dimensionality, the development of practical methods remains an important open problem. For example, can neural networks achieve these rates? Recent concurrent work has provided insight into the use of neural density estimators, see e.g. Bos and Schmidt-Hieber (2023); Cole and Lu (2024); Vandermeulen et al. (2024). Finally, the concept of resilience is a new and unexplored property with significant potential for discovering graphs that yield good estimation properties.

## Acknowledgments and Disclosure of Funding

Robert A. Vandermeulen was supported by German Federal Ministry of Education and Research (BMBF) grants BIFOLD23B and BIFOLD24B. Wai Ming Tai was supported by Singapore AcRF Tier 2 grant MOE-T2EP20122-0001. B.A. was supported by NSF IIS-1956330, NIH R01GM140467, and the Robert H. Topel Faculty Research Fund at the University of Chicago Booth School of Business.

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

# A  Grid Figure

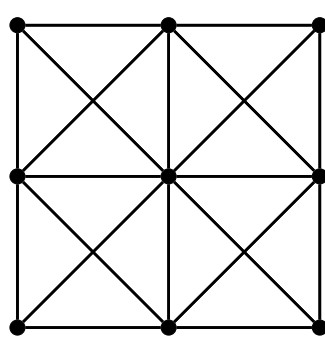 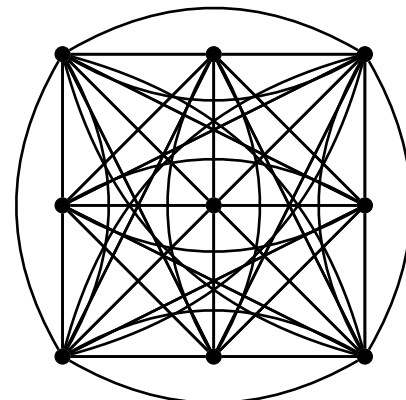

Figure 5: Comparison of $L_{3\times3}$ (left) and $L_{3\times3}^2$ (right).

# B  Proofs on Density Estimation

We begin by introducing some objects and notation that will be essential for proving the main theorems.

Let $\mathbb{N} = \{1, 2, \ldots\}$. For any $m \in \mathbb{N}$, let $[m] \triangleq \{1, 2, \ldots, m\}$. For any $b \in \mathbb{N}$, let $\Delta^b$ be the simplex in $\mathbb{R}^b$, i.e.

$$\Delta^b = \left\{ v \in \mathbb{R}^b : \sum_{i \in [b]} v_i = 1 \text{ and } v_i \geq 0 \text{ for any } i \in [b] \right\}.$$

For any $d, b \in \mathbb{N}$, let $\mathcal{T}_{d,b}$ be the set of probability tensors in $\mathbb{R}^{b^{\times d}}$. This can be thought of as the joint probability tables over $d$ variables with states in $[b]$. For any $d, b \in \mathbb{N}$ and $T \in \mathcal{T}_{d,b}$, we say a random variable $X \sim T$ if

$$P(X = A) = T_A \qquad \text{for all } A \in [b]^d.$$

For any graph $G$ with $d$ vertices, let $\mathcal{T}_b(G) \subseteq \mathcal{T}_{d,b}$, such that, for $T \in \mathcal{T}_b(G)$ and $(X_1, \ldots, X_d) \sim T$, then $X_1, \ldots, X_d$ satisfies the Markov property with respect to the graph $G$. We consider the domain $[0, 1)^d$ and split it into $b$ bins per dimension, i.e. there are $b^d$ bins in total. For any $d, b \in \mathbb{N}$ and multi-index $A \in [b]^d$, let $\Lambda_{d,b,A}$ be the bin indexed at $A$ and $\Lambda_{d,b}$ be the set of all bins, i.e.

$$\Lambda_{d,b,A} = \prod_{i=1}^{d} \left[ \frac{A_i - 1}{b}, \frac{A_i}{b} \right) \qquad \text{and} \qquad \Lambda_{d,b} = \{\Lambda_{d,b,A} \mid A \in [b]^d\}.$$

For any $d, b \in \mathbb{N}$, let $\mathcal{H}_{d,b}$ be the set of all histograms with the bins from $\Lambda_{d,b}$, i.e.

$$\mathcal{H}_{d,b} = \left\{ f : [0, 1)^d \to \mathbb{R} \mid f = \sum_{A \in [b]^d} b^d T_A \cdot \chi_{\Lambda_{d,b,A}} \text{ where } \chi \text{ is the indicator and } T \in \mathcal{T}_{d,b} \right\}.$$

Let $U_{d,b} : \mathcal{T}_{d,b} \to \mathcal{H}_{d,b}$ be the natural $\ell^1 \to L^1$ linear isometry between the spaces.

For any $d \in \mathbb{N}$ and $L > 0$, let $\mathcal{D}_d$ be the set of densities on $[0, 1)^d$ and $\mathcal{D}_{d,L} \subset \mathcal{D}_d$ be those densities which are $L$-Lipschitz continuous.

**Definition B.1.** For $A$, some subset of a $\ell^1$ Euclidean tensor space or $L^1$ function space, we define

$$N(A, \varepsilon) \triangleq \min_{C \subset A} |C| \quad \text{such that, for all } a \in A, \text{ there exists } c \in C \text{ where } \|a - c\|_1 \leq \varepsilon.$$

In this work we will use the following version of the Markov property: for $I, J \subset V$, with $I$ and $J$ not intersecting or adjacent, $X_I \perp\!\!\!\perp X_J \mid X_{V \setminus I \cup J}$. Note that this property is satisfied when the "global" Markov property is satisfied.

**Lemma B.1** (Lemma 3.3.7 from Reiss (1989)). *For probability measures* $\left\| \prod_{i=1}^t \mu_i - \prod_{i=1}^t \nu_i \right\|_1 \leq \sum_{i=1}^t \| \mu_i - \nu_i \|_1$.

**Lemma B.2** (Lemma A.1 from Vandermeulen and Ledent (2021)). *For all $b$ and $0 < \varepsilon \leq 1$,* $N\left( \Delta^b, \varepsilon \right) \leq (2b/\varepsilon)^b$.

**Lemma B.3** (Theorem 3.4 page 7 of Ashtiani et al. (2018), Theorem 3.6 page 54 of Devroye and Lugosi (2001)). *There exists a deterministic algorithm that, given a collection of distributions $p_1, \ldots, p_M$, a parameter $\varepsilon > 0$ and at least $\frac{\log\left(3M^2/\delta\right)}{2\varepsilon^2}$ iid samples from an unknown distribution $p$, outputs an index $j \in [M]$ such that*

$$\| p_j - p \|_1 \leq 3 \min_{i \in [M]} \| p_i - p \|_1 + 4\varepsilon$$

*with probability at least $1 - \frac{\delta}{3}$.*

## B.1 Preliminary Technical Results

This section contains intermediate technical results which will aid in proving more core results.

**Lemma B.4.** *For any $0 < \varepsilon < 1$, any $b, t \in \mathbb{N}$ and any graph $G_i$ for $i \in [t]$, we have*

$$N\left( \mathcal{T}_b(\bigoplus_{i=1}^t G_i), \varepsilon \right) \leq \prod_{i=1}^t N\left( \mathcal{T}_b(G_i), \varepsilon/t \right).$$

*Proof of Lemma B.4.* Let $C_i$ be any $(\varepsilon/t)$-cover for $\mathcal{T}_b(G_i)$ such that $N\left( \mathcal{T}_b(G_i), \varepsilon/t \right) = |C_i|$ for all $i \in [t]$. If we manage to argue that the set

$$C = \{ T_1 \times \cdots \times T_t \mid T_i \in C_i \quad \text{for all } i \in [t] \}$$

is an $\varepsilon$-cover of $\mathcal{T}_b\left( \bigoplus_{i=1}^t G_i \right)$, then we immediately show the statement since

$$N\left( \mathcal{T}_b\left( \bigoplus_{i=1}^t G_i \right), \varepsilon \right) \leq |C| = \prod_{i=1}^t |C_i| = \prod_{i=1}^t N\left( \mathcal{T}_b(G_i), \varepsilon/t \right).$$

Now, to show that $C$ is an $\varepsilon$-cover, we observe that any $T \in \mathcal{T}_b\left( \bigoplus_{i=1}^t G_i \right)$ has the form of $T_1 \times \cdots \times T_t$ where $T_i \in \mathcal{T}_b(G_i)$ for all $i \in [t]$ since we recall that $\bigoplus_{i=1}^t G_i$ is simply a union of the graphs with no edges across different connected components which implies the vertices in different connected components are independent. Since $C_i$ is an $(\varepsilon/t)$-cover of $\mathcal{T}_b(G_i)$, there is a $\widetilde{T}_i \in C_i$ such that $\|\widetilde{T}_i - T_i\|_1 \leq \varepsilon/t$. By taking $\widetilde{T} = \widetilde{T}_1 \times \cdots \times \widetilde{T}_t \in C$, there exists a $\widetilde{T} \in C$ such that, by Lemma B.1, we have

$$\|\widetilde{T} - T\|_1 \leq \sum_{i=1}^t (\varepsilon/t) = \varepsilon.$$

$\square$

**Lemma B.5.** *For any $0 < \alpha < 1$, any $0 < \varepsilon < 1$, any $b \in \mathbb{N}$, any graph $G$ and any vertex $v$ in $G$, we have*

$$N\left( \mathcal{T}_b(G), \varepsilon \right) \leq N\left( \mathcal{T}_{1,b}, \alpha\varepsilon \right) \cdot N\left( \mathcal{T}_b(G \setminus \{v\}), (1-\alpha)\varepsilon \right)^b.$$

*Proof of Lemma B.5.* Without loss of generality, we assume $v = 1$. Suppose the graph $G$ has $d$ vertices. Let $C_0$ be any $(1-\alpha)\varepsilon$-cover of $\mathcal{T}_b(G \setminus \{1\})$ such that $N\left( \mathcal{T}_b(G \setminus \{1\}), (1-\alpha)\varepsilon \right) = |C_0|$

and $C_1$ be any $\alpha\varepsilon$-cover of $\mathcal{T}_{1,b}$ such that $N\left(\mathcal{T}_{1,b}, \alpha\varepsilon\right) = |C_1|$. We now construct a set $C \in \mathcal{T}_b(G)$ as follows.

$$C = \{T \mid T_A = \delta_{A_1} \cdot T^{(A_1)}_{(A_2,\ldots,A_d)} \quad \text{for any } A \in [b]^d \text{ where } T^{(1)}, \ldots, T^{(b)} \in C_0 \text{ and } \delta \in C_1\}.$$

If we manage to argue that the set $C$ is an $\varepsilon$-cover of $\mathcal{T}_b(G)$, then we immediately show the statement since

$$N\left(\mathcal{T}_b(G), \varepsilon\right) \leq |C| = |C_1| \cdot |C_0|^b = N\left(\mathcal{T}_{1,b}, \alpha\varepsilon\right) \cdot N\left(\mathcal{T}_b\left(G \setminus \{1\}\right), (1-\alpha)\varepsilon\right)^b.$$

Now, to show that $C$ is an $\varepsilon$-cover, we observe that, for any $T \in \mathcal{T}_b(G)$ and $A \in [b]^d$, we can express

$$T_A = \delta_{A_1} \cdot T^{(A_1)}_{(A_2,\ldots,A_d)} \quad \text{for some } T^{(1)}, \ldots, T^{(b)} \in \mathcal{T}_b\left(G \setminus \{1\}\right) \text{ and } \delta \in \mathcal{T}_{1,b}.$$

Since $C_0$ (resp. $C_1$) is an $(1-\alpha)\varepsilon$-cover of $\mathcal{T}_b\left(G \setminus \{1\}\right)$ (resp. an $\alpha\varepsilon$-cover of $\mathcal{T}_{1,b}$), there exist $\widetilde{T}^{(1)}, \ldots, \widetilde{T}^{(b)} \in C_0$ and $\widetilde{\delta} \in C_1$ such that

$$\|\widetilde{T}^{(i)} - T^{(i)}\|_1 \leq (1-\alpha)\varepsilon \quad \text{for all } i \in [b] \quad \text{and} \quad \|\widetilde{\delta} - \delta\|_1 \leq \alpha\varepsilon. \tag{4}$$

Then, we take $\widetilde{T} \in C$ where

$$\widetilde{T}_A = \widetilde{\delta}_{A_1} \cdot \widetilde{T}^{(A_1)}_{(A_2,\ldots,A_d)} \quad \text{for any } A \in [b]^d$$

and we have

$$\|\widetilde{T} - T\|_1 = \sum_{A_1=1}^{b} \|\widetilde{\delta}_{A_1} \cdot \widetilde{T}^{(A_1)} - \delta_{A_1} \cdot T^{(A_1)}\|_1$$

$$\leq \sum_{A_1=1}^{b} \|\widetilde{\delta}_{A_1} \cdot \widetilde{T}^{(A_1)} - \delta_{A_1} \cdot \widetilde{T}^{(A_1)}\|_1 + \sum_{A_1=1}^{b} \|\delta_{A_1} \cdot \widetilde{T}^{(A_1)} - \delta_{A_1} \cdot T^{(A_1)}\|_1.$$

By using (4), we further bound each term

$$\sum_{A_1=1}^{b} \|\widetilde{\delta}_{A_1} \cdot \widetilde{T}^{(A_1)} - \delta_{A_1} \cdot \widetilde{T}^{(A_1)}\|_1 = \sum_{A_1=1}^{b} |\widetilde{\delta}_{A_1} - \delta_{A_1}| \cdot \|\widetilde{T}^{(A_1)}\|_1 = \|\widetilde{\delta} - \delta\|_1 \leq \alpha\varepsilon \quad \text{and}$$

$$\sum_{A_1=1}^{b} \|\delta_{A_1} \cdot \widetilde{T}^{(A_1)} - \delta_{A_1} \cdot T^{(A_1)}\|_1 = \sum_{A_1=1}^{b} \delta_{A_1} \cdot \|\widetilde{T}^{(A_1)} - T^{(A_1)}\|_1 \leq \sum_{A_1=1}^{b} \delta_{A_1} \cdot (1-\alpha)\varepsilon = (1-\alpha)\varepsilon.$$

Combining these two inequalities, we conclude that, for any $T \in \mathcal{T}_b(G)$, there exists a $\widetilde{T} \in C$ such that $\|\widetilde{T} - T\|_1 \leq \alpha\varepsilon + (1-\alpha)\varepsilon = \varepsilon$. $\qquad\square$

## B.2 Controlling Bias and Variance

Like many the analysis of many estimators this proof has two pain parts, analysis of the bias, and the analysis of the variance.

### B.2.1 Variance

To control the variance we will use the follow bound on covering numbers

**Proposition B.6.** *For any $b \in \mathbb{N}$, any $0 < \varepsilon < 1$ and any graph $G$ whose number of vertices is $d$ and resilience is $r$, we have*

$$\log N\left(\mathcal{T}_b(G), \varepsilon\right) \leq db^r \log\left(\frac{2d^{r+1}b}{\varepsilon}\right).$$

*Proof of Proposition B.6.* We first assume that $G$ is a connected graph and prove that

$$\log N\left(\mathcal{T}_b(G), \varepsilon\right) \leq db^r \log\left(\frac{2d^r b}{\varepsilon}\right). \tag{5}$$

Note that the exponent in the factor $d^r$ is $r$ as opposed to $r+1$ in the original statment. We will prove the statement by induction on $r$.

**Base case $r = 1$:** Note that the graph $G$ is a graph with only one vertex. By Lemma B.2, we have

$$\log N(\mathcal{T}_b(G), \varepsilon) \leq \log N(\Delta^b, \varepsilon) \leq b \log \left( \frac{2b}{\varepsilon} \right) = 1 \cdot b^1 \log \left( \frac{2 \cdot 1^1 \cdot b}{\varepsilon} \right),$$

thereby satisfying (5).

**Inductive step $r > 1$:** Let $v$ be the vertex selected in the first disintegration step of the $r$-disintegration and $G_1, \ldots, G_t$ be the connected components after removing $v$ from $G$. Also, for all $i \in [t]$, let $r_i = r(G_i)$ and $d_i$ be the number of vertices in $G_i$. By Lemma B.5, we first have

$$\log N\left(\mathcal{T}_b(G), \varepsilon\right) \leq \log N\left(\mathcal{T}_{1,b}, \frac{\varepsilon}{t+1}\right) + b \cdot \log N\left(\mathcal{T}_b(G \setminus \{v\}), \frac{t\varepsilon}{t+1}\right)$$

$$\leq \underbrace{b \log \left(\frac{2(t+1)b}{\varepsilon}\right)}_{\text{by Lemma B.2}} + b \cdot \log N\left(\mathcal{T}_b(G \setminus \{v\}), \frac{t\varepsilon}{t+1}\right). \tag{6}$$

By Lemma B.4, we further have

$$\log N\left(\mathcal{T}_b(G \setminus \{v\}), \frac{t\varepsilon}{t+1}\right) \leq \sum_{i=1}^{t} \log N\left(\mathcal{T}_b(G_i), \frac{\varepsilon}{t+1}\right). \tag{7}$$

Note that the resilience of each $G_i$ is at most $r - 1$ (i.e. $r_i \leq r - 1$) and hence we invoke the inductive assumption. We have

$$\log N\left(\mathcal{T}_b(G_i), \frac{\varepsilon}{t+1}\right) \leq d_i \cdot b^{r_i} \cdot \log \left(\frac{2(t+1)d_i^{r_i}b}{\varepsilon}\right) \qquad \text{by the induction assumption}$$

$$\leq d_i \cdot b^{r-1} \cdot \log \left(\frac{2d^r b}{\varepsilon}\right) \qquad \text{since } r_i \leq r - 1 \text{ and } t+1, d_i \leq d. \tag{8}$$

By plugging (8) into (7) and (6), we have

$$\log N\left(\mathcal{T}_b(G), \varepsilon\right) \leq b \log \left(\frac{2(t+1)b}{\varepsilon}\right) + b \cdot \sum_{i=1}^{t} d_i \cdot b^{r-1} \cdot \log \left(\frac{2d^r b}{\varepsilon}\right)$$

$$\leq b^r \log \left(\frac{2d^r b}{\varepsilon}\right) \cdot \left(1 + \sum_{i=1}^{t} d_i\right) \qquad \text{since } t + 1 \leq d^r$$

$$= db^r \log \left(\frac{2d^r b}{\varepsilon}\right) \qquad \text{since } 1 + \sum_{i=1}^{t} d_i = d.$$

Now, we remove the assumption of $G$ being connected. Suppose $G$ has $t$ connected components, $G_1, \ldots, G_t$ whose number of vertices is $d_i$ and resilience is $r_i$. We have

$$\log N(\mathcal{T}_b(G), \varepsilon) \leq \sum_{i=1}^{t} \log N\left(\mathcal{T}_b(G_i), \frac{\varepsilon}{t}\right) \qquad \text{by Lemma B.4}$$

$$\leq \sum_{i=1}^{t} d_i b^{r_i} \log \left(\frac{2td_i^{r_i}b}{\varepsilon}\right) \qquad \text{from the case of } G \text{ being connected}$$

$$\leq db^r \log \left(\frac{2d^{r+1}b}{\varepsilon}\right) \qquad \text{since } d_i \leq \sum_{i=1}^{t} d_i = d, r_i \leq r \text{ and } t \leq d.$$

$\square$

### B.2.2 Analysis of Bias

**Theorem B.7.** *For any $d, b \in \mathbb{N}$, any $L > 0$, any graph $G$ and any $p \in \mathcal{D}_L(G)$, we have*

$$\min_{p' \in U_{d,b}(\mathcal{T}_b(G))} \|p - p'\|_1 \leq \sqrt{d}L/b.$$

*Proof of Theorem B.7.* For any $d, b \in \mathbb{N}$ and $A \in [b]^d$, recall that $\Lambda_{d,b,A} = \prod_{i=1}^{d} \left[ \frac{A_i-1}{b}, \frac{A_i}{b} \right)$. Let $\lambda_A$ be the centroid of $\Lambda_{d,b,A}$. We define

$$\widetilde{p}' = \sum_{A \in [b]^d} p(\lambda_A) \chi_{\Lambda_{d,b,A}} \quad \text{and} \quad \widetilde{p} = \widetilde{p}'/Z \quad \text{if } Z \neq 0 \tag{9}$$

where $\chi_{\Lambda_{d,b,A}}$ is the indicator, i.e. $\chi_{\Lambda_{d,b,A}}(x) = \begin{cases} 1 & \text{if } x \in \Lambda_{d,b,A} \\ 0 & \text{if } x \notin \Lambda_{d,b,A} \end{cases}$ and $Z$ is the normalizing factor,

i.e. $Z = \int_{x \in [0,1)^d} \widetilde{p}'(x) dx$.

Since the construction requires $Z \neq 0$, we now first show that when $Z = 0$ the statement holds trivially. When $Z = 0$, we have

$$p(\lambda_A) = 0 \quad \text{for all } A \in [b]^d$$

which implies

$$p(x) \leq L \cdot \|x - \lambda_A\|_2 \leq L\sqrt{d}/(2b) \quad \text{since } p \in \mathcal{D}_L(G).$$

Hence, we further have

$$1 = \int_{x \in [0,1)^d} p(x) dx = \sum_{A \in [b]^d} \int_{x \in \Lambda_{d,b,A}} p(x) dx \leq \sum_{A \in [b]^d} \int_{x \in \Lambda_{d,b,A}} L\sqrt{d}/(2b) dx = L\sqrt{d}/(2b).$$

Namely, $L\sqrt{d}/b \geq 2$ and therefore the statement holds for any $p' \in U_{d,b}(\mathcal{T}_b(G))$.

Now, we assume that $Z \neq 0$. We will prove the statement by arguing 1) $\widetilde{p} \in U_{d,b}(\mathcal{T}_b(G))$ and 2) $\|p - \widetilde{p}\|_1 \leq \sqrt{d}L/b$.

**To prove 1)** $\widetilde{p} \in U_{d,b}(\mathcal{T}_b(G))$**:**   For any partition $V_0 \cup V_1 \cup V_2$ of the vertex set of $G$ such that $V_0$ separates $V_1$ and $V_2$, we would like to show, for any $x^* \in [0,1)^d$ (without loss of generality, we reorder the indices such that $x = (x_{V_1}, x_{V_2}, x_{V_0})$)

$$\frac{\widetilde{p}(x^*_{V_1}, x^*_{V_2}, x^*_{V_0})}{\widetilde{p}(:,:,x^*_{V_0})} = \frac{\widetilde{p}(x^*_{V_1},:,x^*_{V_0})}{\widetilde{p}(:,:,x^*_{V_0})} \cdot \frac{\widetilde{p}(:,x^*_{V_2},x^*_{V_0})}{\widetilde{p}(:,:,x^*_{V_0})} \quad \text{if } \widetilde{p}(:,:,x^*_{V_0}) \neq 0$$

where we use $:$ to indicate integrating with respect to the corresponding variables, i.e.

$$\widetilde{p}(:,:,x^*_{V_0}) = \int_{(x_{V_1}, x_{V_2}) \in [0,1)^{d_1+d_2}} \widetilde{p}(x_{V_1}, x_{V_2}, x^*_{V_0}) d(x_{V_1}, x_{V_2})$$

$$\widetilde{p}(:,x^*_{V_2},x^*_{V_0}) = \int_{x_{V_1} \in [0,1)^{d_1}} \widetilde{p}(x_{V_1}, x^*_{V_2}, x^*_{V_0}) dx_{V_1}$$

$$\widetilde{p}(x^*_{V_1},:,x^*_{V_0}) = \int_{x_{V_2} \in [0,1)^{d_2}} \widetilde{p}(x^*_{V_1}, x_{V_2}, x^*_{V_0}) dx_{V_2}$$

and $d_1$ (resp. $d_2$) is the size of $V_1$ (resp. $V_2$). It is equivalent to show

$$\widetilde{p}(x^*_{V_1}, x^*_{V_2}, x^*_{V_0}) \cdot \widetilde{p}(:,:,x^*_{V_0}) = \widetilde{p}(x^*_{V_1},:,x^*_{V_0}) \cdot \widetilde{p}(:,x^*_{V_2},x^*_{V_0}). \tag{10}$$

We now analyze each of $\widetilde{p}(x^*_{V_1}, x^*_{V_2}, x^*_{V_0}), \widetilde{p}(:,:,x^*_{V_0}), \widetilde{p}(x^*_{V_1},:,x^*_{V_0}), \widetilde{p}(:,x^*_{V_2},x^*_{V_0})$. Let $\lambda$ (resp. $\lambda^*$) be the closest centroid to $x$ for any $x \in [0,1)^d$ (resp. $x^*$). For $\widetilde{p}'(x^*_{V_1}, x^*_{V_2}, x^*_{V_0})$, we have

$$\begin{aligned}
\widetilde{p}(x^*_{V_1}, x^*_{V_2}, x^*_{V_0}) &= \frac{1}{Z} p(\lambda^*_{V_1}, \lambda^*_{V_2}, \lambda^*_{V_0}) \\
&= \frac{p(\lambda^*_{V_1}, :, \lambda^*_{V_0}) \cdot p(:, \lambda^*_{V_2}, \lambda^*_{V_0})}{Z p(:,:,\lambda^*_{V_0})} \quad \text{since } p \in \mathcal{D}_L(G) \text{ and } V_0 \text{ separates } V_1 \text{ and } V_2.
\end{aligned} \tag{11}$$

Here, recall that we use : to indicate integrating with respect to the corresponding variables, i.e., for any centroid $\lambda$ including $\lambda^*$,

$$p(\lambda_{V_1}, : , \lambda_{V_0}) = \int_{x_{V_2} \in [0,1)^{d_2}} p(\lambda_{V_1}, x_{V_2}, \lambda_{V_0}) dx_{V_2}$$

$$p(: , \lambda_{V_2}, \lambda_{V_0}) = \int_{x_{V_1} \in [0,1)^{d_1}} p(x_{V_1}, \lambda_{V_2}, \lambda_{V_0}) dx_{V_1}$$

$$p(: , : , \lambda_{V_0}) = \int_{(x_{V_1}, x_{V_2}) \in [0,1)^{d_1+d_2}} p(x_{V_1}, x_{V_2}, \lambda_{V_0}) d(x_{V_1}, x_{V_2}).$$

For $\widetilde{p}'(: , : , x_{V_0}^*)$, we have

$$\widetilde{p}(: , : , x_{V_0}^*) = \int_{(x_{V_1}, x_{V_2}) \in [0,1)^{d_1+d_2}} \widetilde{p}(x_{V_1}, x_{V_2}, x_{V_0}^*) d(x_{V_1}, x_{V_2})$$

$$= \frac{1}{Z} \int_{(x_{V_1}, x_{V_2}) \in [0,1)^{d_1+d_2}} \widetilde{p}'(x_{V_1}, x_{V_2}, x_{V_0}^*) d(x_{V_1}, x_{V_2})$$

Since, for any $A \in [b]^d$, $\widetilde{p}'(x_{V_1}, x_{V_2}, x_{V_0}^*) = p(\lambda_{V_1}, \lambda_{V_2}, \lambda_{V_0}^*)$ for all $(x_{V_1}, x_{V_2}, x_{V_0}^*) \in \Lambda_{d,b,A}$ by (9), we have

$$\int_{(x_{V_1}, x_{V_2}) \in \Lambda_{d,b,A}} \widetilde{p}'(x_{V_1}, x_{V_2}, x_{V_0}^*) d(x_{V_1}, x_{V_2}) = \frac{1}{b^{d_1+d_2}} p(\lambda_{V_1}, \lambda_{V_2}, \lambda_{V_0}^*).$$

By plugging it into the equation for $\widetilde{p}(: , : , x_{V_0}^*)$, we have

$$\widetilde{p}(: , : , x_{V_0}^*) = \sum_{\text{centroid } \lambda \,:\, \lambda_{V_0} = \lambda_{V_0}^*} \frac{1}{Zb^{d_1+d_2}} p(\lambda_{V_1}, \lambda_{V_2}, \lambda_{V_0}^*)$$

$$= \sum_{\text{centroid } \lambda \,:\, \lambda_{V_0} = \lambda_{V_0}^*} \frac{p(\lambda_{V_1}, : , \lambda_{V_0}^*) \cdot p(: , \lambda_{V_2}, \lambda_{V_0}^*)}{Zb^{d_1+d_2} p(: , : , \lambda_{V_0}^*)}. \tag{12}$$

For $\widetilde{p}'(x_{V_1}^*, : , x_{V_0}^*)$, we have

$$\widetilde{p}(x_{V_1}^*, : , x_{V_0}^*) = \sum_{\text{centroid } \lambda \,:\, (\lambda_{V_1}, \lambda_{V_0}) = (\lambda_{V_1}^*, \lambda_{V_0}^*)} \frac{1}{Zb^{d_2}} p(\lambda_{V_1}^*, \lambda_{V_2}, \lambda_{V_0}^*)$$

$$= \sum_{\text{centroid } \lambda \,:\, (\lambda_{V_1}, \lambda_{V_0}) = (\lambda_{V_1}^*, \lambda_{V_0}^*)} \frac{p(\lambda_{V_1}^*, : , \lambda_{V_0}^*) \cdot p(: , \lambda_{V_2}, \lambda_{V_0}^*)}{Zb^{d_2} p(: , : , \lambda_{V_0}^*)}. \tag{13}$$

Similarly, for $\widetilde{p}'(: , x_{V_2}^*, x_{V_0}^*)$, we have

$$\widetilde{p}(: , x_{V_2}^*, x_{V_0}^*) = \sum_{\text{centroid } \lambda \,:\, (\lambda_{V_2}, \lambda_{V_0}) = (\lambda_{V_2}^*, \lambda_{V_0}^*)} \frac{p(: , \lambda_{V_2}^*, \lambda_{V_0}^*) \cdot p(\lambda_{V_1}, : , \lambda_{V_0}^*)}{Zb^{d_1} p(: , : , \lambda_{V_0}^*)}. \tag{14}$$

By using (11), (12), (13) and (14), we can conclude (10).

**To prove 2)** $\|p - \widetilde{p}\|_1 \le \sqrt{d}L/b$: We first express

$$\|p - \widetilde{p}\|_1 \le \|p - \widetilde{p}'\|_1 + \|\widetilde{p}' - \widetilde{p}\|_1 \quad \text{by triangle inequality.} \tag{15}$$

Note that $\widetilde{p}'$ may not be a probability density.

For the first term $\|p - \widetilde{p}'\|_1$, we have

$$\|p - \widetilde{p}'\|_1 = \int_{x \in [0,1)^d} |p(x) - \widetilde{p}'(x)| dx = \sum_{A \in [b]^d} \int_{x \in \Lambda_{d,b,A}} |p(x) - \widetilde{p}'(x)| dx. \tag{16}$$

For any $x \in \Lambda_{d,b,A}$, we have $\widetilde{p}'(x) = p(\lambda_A)$ by the definition of $\widetilde{p}'$. Also, since $p \in \mathcal{D}_L(G)$, $p$ satisfies the $L$-Lipschitz condition which implies

$$|p(x) - \widetilde{p}'(x)| = |p(x) - p(\lambda_A)| \le L \cdot \|x - \lambda_A\|_2 \le L\sqrt{d}/(2b). \tag{17}$$

By plugging (17) into (16), we have

$$\|p - \widetilde{p}'\|_1 \leq \sum_{A \in [b]^d} \int_{x \in \Lambda_{d,b,A}} L\sqrt{d}/(2b) dx = L\sqrt{d}/(2b). \qquad (18)$$

For the second term $\|\widetilde{p}' - \widetilde{p}\|_1$, we observe that $\widetilde{p}$ is simply a scaled version of $\widetilde{p}'$ by a factor of $1/Z$. Namely, we have

$$\|\widetilde{p}'\|_1 = Z \quad \text{and} \quad \|\widetilde{p}' - \widetilde{p}\|_1 = |Z - 1| \quad \text{since } \|\widetilde{p}\|_1 = 1.$$

By (18) and the fact of $\|p\|_1 = 1$, we further have

$$|Z - 1| = |\|\widetilde{p}'\|_1 - 1| \leq \|p - \widetilde{p}'\|_1 \leq L\sqrt{d}/(2b) \quad \text{which implies} \quad \|\widetilde{p}' - \widetilde{p}\|_1 \leq L\sqrt{d}/(2b). \qquad (19)$$

By plugging (18) and (19) into (15), we have

$$\|p - \widetilde{p}\|_1 \leq L\sqrt{d}/(2b) + L\sqrt{d}/(2b) = L\sqrt{d}/b.$$

$\square$

## B.3   Proof of Main Theorems

With the previous results established we can now prove the core sample complexity results of this work.

*Proof of Theorem 3.1.* Recall that Lemma B.3 states the following. There is a deterministic algorithm that, given a collection of $M$ distributions $C = \{p_1, \ldots, p_M\}$, any $0 < \varepsilon < 1$ and at least $\frac{\log(3M^2/\delta)}{2\varepsilon^2}$ i.i.d. samples drawn from an unknown distribution $p$, outputs an index $j \in [M]$ such that

$$\|p_j - p\|_1 \leq 3 \min_{i \in [M]} \|p_i - p\|_1 + 4\varepsilon \quad \text{with probability at least } 1 - \frac{\delta}{3}.$$

We will use the algorithm from Lemma B.3 by taking the collection $C$ to be the $\varepsilon$-cover for $\mathcal{T}_b(G)$. Here, we determine $b \in \mathbb{N}$ later. By Proposition B.6, we have

$$\log M = \log|C| \leq db^r \log(\frac{2d^{r+1}b}{\varepsilon})$$

and, by Theorem B.7, there exists a $p' \in U_{d,b}(\mathcal{T}_b(G))$ such that

$$\|p - p'\|_1 \leq \frac{\sqrt{d}L}{b}$$

which implies there exists a $p'' \in C$ such that

$$\|p - p''\|_1 \leq \|p - p'\|_1 + \|p' - p''\|_1 \leq \frac{\sqrt{d}L}{b} + \varepsilon.$$

Namely, whenever $n \geq \frac{db^r \log(\frac{2d^{r+1}b}{\varepsilon})}{\varepsilon^2} + \frac{\log(3/\delta)}{2\varepsilon^2}$ i.i.d. samples are drawn from $p$, the distribution $q$ returned by the algorithm in Lemma B.3 satisfies

$$\|q - p\|_1 \leq 3\left(\frac{\sqrt{d}L}{b} + \varepsilon\right) + 4\varepsilon = \frac{3\sqrt{d}L}{b} + 7\varepsilon \quad \text{with probability at least } 1 - \frac{\delta}{3}.$$

Now, by picking $b = \Theta(\frac{\sqrt{d}L}{\varepsilon})$, the desired result follows. $\square$

*Proof of Theorem 3.2.* The proof is the same as in Theorem 3.1 except that we need to another $\varepsilon$-cover for $\bigcup_{G \in \mathcal{G}} \mathcal{T}_b(G)$. For any $G \in \mathcal{G}$, let $C_G$ be the $\varepsilon$-cover for $\mathcal{T}_b(G)$. By taking $\bigcup_{G \in \mathcal{G}} C_G$ to be the $\varepsilon$-cover for $\bigcup_{G \in \mathcal{G}} \mathcal{T}_b(G)$, we have

$$\log M = \log|C| \leq \log\Big(\sum_{G \in \mathcal{G}} |C_G|\Big).$$

By Proposition B.6, each of $|C_G|$ is bounded above by $(\frac{2d^{r+1}b}{\varepsilon})^{db^r}$ and, by considering all possible graphs, $|\mathcal{G}|$ is bounded above by $2^{d^2}$. In particular, when $r = 1$, we have $|\mathcal{G}| = 1$. Hence, we have

$$\log M \leq \begin{cases} db^r \log(\frac{2d^{r+1}b}{\varepsilon}) + d^2 & \text{if } r > 1 \\ db \log(\frac{2d^2b}{\varepsilon}) & \text{if } r = 1 \end{cases}$$

and we can conclude the desired result by following the rest of the proof in Theorem 3.1. $\square$

# C  Proofs on Graph Resilience

For proofs in this section it will be implicit that graphs with a subscript, $G_i$, are equal to $(V_i, E_i)$.

For proofs in this section we will introduce a useful concept we term a *pre-disintegration of a graph*.

**Definition C.1.** For any graph $G = (V, E)$, we define a *pre-disintegration* of $G$ to be a tuple of subsets of $V$ which satisfies the properties 1, 2, and, 4 in Definition 3.1 of disintegration, noting that some entries of a pre-disintegration may be the empty set, including the first entry.

In other words, the only difference between a pre-disintegration and a disintegration is that, in each step, we do not have to select a vertex in every connected component. The *length of a pre-disintegration*, $D$, is defined to be the largest $r$ such that $D_r$ is nonempty and we call such pre-disintegration a $r$-pre-disintegration. We now have the following lemma on pre-disintegrations.

**Lemma C.1.** *For any graph $G$, if there is a $\ell$-pre-disintegration of $G$, then $r(G) \leq \ell$.*

*Proof of Lemma C.1.* Let $\mathcal{D}$ be the set of all pre-disintegration of $G$. For any $D \in \mathcal{D}$ and any vertex $v \in V$ where $V$ is the vertex set of $G$, we define $D^{-1}$ to be a function from $V$ to $\mathbb{N}$ such that $D^{-1}(v)$ is the index $i$ where $v \in D_i$. We will now define a partial order on $\mathcal{D}$. For any $D$ and $D'$ in $\mathcal{D}$, we say $D \leq D'$ if $D^{-1}(v) \leq D'^{-1}(v)$ for all $v \in V$. Namely, every vertex $v$ appears no later in $D$ than in $D'$. From the assumption, there exists a $\ell$-pre-disintegration of $G$, $\widehat{D}$. Similarly to the definition of the length of a pre-disintegration, trailing empty sets are simply ignored to satisfy the antisymmetry property for a partial order. Hence, the set $\{D \in \mathcal{D} \mid D \leq \widehat{D}\}$ is finite and there must exist a minimal element, $D^*$. By removing trailing empty sets from $D^*$, the length of $D^*$ is no larger than $\ell$.

Now, we need to show $D^*$ is a disintegration. Since $D^*$ is a pre-disintegration, it satisfies the properties 1, 2, and, 4 in Definition 3.1. Namely, we need to check the properties 3 and 5. For 3, we prove it by contradiction. When $D^*$ violates 3, there is a connected component $G'$ of $G$ such that no $v \in V'$, where $V'$ is the vertex set of $G'$, is in $D_1^*$. Let $v'$ be $\arg\min_{v \in V'} D^{*-1}(v)$ and $i'$ be $D^{*-1}(v')$, i.e. the earliest vertex in $V'$ appears in $D^*$. We construct a new pre-disintegration $\widehat{D}^*$ by setting $\widehat{D}^* = D^*$ except that $\widehat{D}_1^* = D_1^* \cup \{v'\}$ and $\widehat{D}_{i'}^* = D_{i'}^* \setminus \{v'\}$, i.e. we move $v'$ to the first step. Hence, we have $\widehat{D}^* \leq D^*$ and $\widehat{D}^* \neq D^*$ which contradicts the minimality of $D^*$. We can perform the identical argument to show 5. $\square$

Equipped with Lemma C.1, we can now commence with the proofs from the main text.

*Proof of Lemma 4.1.* We will assume $V_1, \ldots, V_m$ contain distinct elements for convenience. We now prove the statement by showing the following two inequalities: 1) $\max_{i \in [m]} r(G_i) \geq r(\bigoplus_{i \in [m]} G_i)$ and 2) $\max_{i \in [m]} r(G_i) \leq r(\bigoplus_{i \in [m]} G_i)$.

**For 1)** $\max_{i \in [m]} r(G_i) \geq r(\bigoplus_{i \in [m]} G_i)$**:**  Let $r$ be the maximum of the graph resilience of $G_i$, i.e. $r = \max_{i \in [m]} r(G_i)$, and $D^{(i)}$ be a $r(G_i)$-disintegration for $i \in [m]$. We construct a $r$-tuple $D$ by setting the $j$-th step to be $D_j = \bigcup_{i \in [m]} D_j^{(i)}$. It is easy to check that $D$ satisfies Definition 3.1 and hence $D$ is a $r$-disintegration of $\bigoplus_{i \in [m]} G_i$. By Definition 3.2, the inequality $r \geq r(\bigoplus_{i \in [m]} G_i)$ follows.

**For 2)** $\max_{i \in [m]} r(G_i) \leq r(\bigoplus_{i \in [m]} G_i)$**:**  Let $r$ be the graph resilience of $\bigoplus_{i \in [m]} G_i$, i.e. $r = r(\bigoplus_{i \in [m]} G_i)$, and $D$ be a $r$-disintegration of $\bigoplus_{i \in [m]} G_i$. We construct $m$ tuples $D^{(1)}, \ldots, D^{(m)}$ by setting

$$D_i^{(j)} = \{v \in V \mid v \in V_j \cap D_i\} \quad \text{for all } j \in [m] \text{ and } i \in [r].$$

It is easy to check that, for each $j \in [m]$, $D^{(j)}$ satisfies Definition C.1 and hence $D^{(j)}$ is a $r$-pre-disintegration of $G_j$. By Definition 3.2 and Lemma C.1, we have $r(G_i) \leq r$ for all $i \in [m]$ and hence the inequality $\max_{i \in [m]} r(G_i) \leq r$ follows. $\square$

*Proof of Lemma 4.2.* We will prove for the case where $|V'| = 1$. Note that the lemma statement follows from repeated application of this case.

We will proceed by contradiction. Suppose there exists $G$ and $v$ such $r(G) \geq r(G \setminus \{v\}) + 2$. Let $\widetilde{D}$ be a $r(G \setminus \{v\})$-disintegration of $G \setminus \{v\}$. If we define a tuple $D$ with $D_1 = v$ and $D_{i+1} = \widetilde{D}_i$ for all $i$, it is clear that $D$ is a $(r(G \setminus \{v\}) + 1)$-pre-disintegration of $G$. The contradiction follows from the application of Lemma C.1. $\square$

*Proof of Lemma 4.3.* We will prove for the case where $|E'| = 1$, the lemma follows from repeated application of this case. Without loss of generality let $E' = (1, 2)$. Let $D$ be an $r$-disintegration of $G$. Let $\widetilde{D}$ be a $(r + 1)$-tuple with $\widetilde{D}_1 = \{1\}$ and $\widetilde{D}_i = D_{i-1} \setminus \{1\}$ for $i \in \{2, \ldots, r+1\}$. Then $\widetilde{D}$ is an pre-disintegration of $G'$ with length $r + 1$. This case follows from Lemma C.1. $\square$

*Proof of Lemma 4.4.* Let $D$ be a $r(G)$-disintegration. Let $\widetilde{D}$ be the sequence of subsets of $V'$ when one removes the elements of in $V \setminus V'$ from $D$. Clearly, $\widetilde{D}$ is a pre-disintegration of $G'$, with length at most $r(G)$. The lemma follows from application of Lemma C.1. $\square$

*Proof of Lemma 4.6.* It is clear that when $G$ with $d$ vertices is a complete graph that $r(G) = d$. For the reverse direction, suppose that $G$ is a graph with $V = [d]$ with no edge between $d - 1$ and $d$. We can define a pre-disintegration where $D_i = \{i\}$ for $i \in [d - 2]$ and $D_{d-1} = \{d - 1, d\}$. The lemma follows from application of Lemma C.1. $\square$

*Proof of Lemma 4.7.* When $E = \emptyset$ clearly $D_1 = V$ is a disintegration.

If $E \neq \emptyset$ then there must exist two vertices in some component so $D_1 = V$ does not satisfy the component property of a disintegration, thus $D_2 \neq \emptyset$ for any disintegration of $G$. $\square$

*Proof of Lemma 4.8.* One can simply choose the center vertex as the first entry of a disintegration and the remaining vertices as the rest. $\square$

*Proof of Lemma 4.9.* We construct a $r$-tuple $D$ as follows. For all $i \in [r]$, set $D_i$ to be the set of all tree vertices at the $i$-th level in $G$. Here, we define the root node to be at the first level. It is easy to check that $D$ a $r$-disintegration. By Definition 3.2, we have $r(G) \leq r$. $\square$

*Proof of Lemma 4.10.* Let $V$ be the vertex set. For any vertex $v \in V$, let $\alpha_v$ be the maximum number of vertices in the connected components after removing $v$. We would like to show that there exists a vertex $v^*$ such that $\alpha_{v^*} \leq \lfloor \frac{d}{2} \rfloor$. We prove it by contradiction. Let $v'$ be $\arg\min_{v \in V} \alpha_v$ and $V_1, \ldots, V_t$ be the vertex sets of the connected components after removing $v'$. We have $\alpha_{v'} > \lfloor \frac{d}{2} \rfloor$ which means that there is a connected component whose number of vertices is strictly larger than $\lfloor \frac{d}{2} \rfloor$. Note that there is only one such component since the sum of the number of vertices of other components is strictly less than $d - 1 - \lfloor \frac{d}{2} \rfloor$. WLOG, let $V_1$ be the vertex set of that component. Let $v''$ be the vertex in $V_1$ sharing an edge with $v'$. We want to argue that $\alpha_{v''} < \alpha_{v'}$ which contradicts the definition of $v'$. To see this, we examine the number of vertices in the connected components after removing $v''$. We use the fact that $G$ is a tree. For the component whose vertex set is $\cup_{i=2}^{t} V_i \cup \{v'\}$, the size is strictly less than $(d - 1 - \lfloor \frac{d}{2} \rfloor) + 1 = d - \lfloor \frac{d}{2} \rfloor \leq \alpha_{v'}$. For any component inside $V_1$, the size is less than $\lfloor \frac{d}{2} \rfloor - 1 < \alpha_{v'}$. Namely, we have $\alpha_{v''} < \alpha_{v'}$. Therefore, there exists a vertex $v^*$ such that $\alpha_{v^*} \leq \lfloor \frac{d}{2} \rfloor$.

Now, we remove $v^*$ in the first step and all the connected components, $G_1, \ldots, G_t$, has at most $\lfloor \frac{d}{2} \rfloor$ vertices. By induction, we have

$$
\begin{aligned}
r(G) &\leq r(G \setminus \{v^*\}) + 1 \quad \text{by Lemma 4.2} \\
&= r\Big(\bigoplus_{i \in [t]} G_i\Big) + 1 \\
&= \max_{i \in [t]} r(G_i) + 1 \quad \text{by Lemma 4.1} \\
&\leq \big(\log_2(\lfloor \tfrac{d}{2} \rfloor) + 1\big) + 1 \quad \text{by the inductive assumption} \\
&= \log_2(d) + 1.
\end{aligned}
$$

$\square$

*Proof of Proposition 4.11.* We will prove the statement by induction on $s$.

**Base case $s = 1$:** When $s = 1$, we observe that $L^t_{t(2^1-1)} = L^t_t$ is a complete graph with $t$ vertices. By Lemma 4.6, we have $r(L^t_t) = t \leq 1 \cdot t$.

**Inductive step $s > 1$:** Let $V' \subset V$ be $\{t(2^{s-1} - 1) + i \mid i \in [t]\}$. Note that $|V'| = t$. By Lemma 4.2, we have

$$
r(L^t_{t(2^s-1)}) \leq r(L^t_{t(2^s-1)} \setminus V') + t.
$$

In other words, we first remove the "middle" $t$ vertices from the graph. By the definition of path graphs (Definition 4.2), it is easy to check that there is no edge between $i$ and $j$ for $i = 1, \ldots, t(2^{s-1} - 1)$ and $j = t(2^{s-1} - 1) + t + 1, \ldots, t(2^s - 1)$. Therefore, there are two connected components and clearly each of them is isomorphic to $L^t_{t(2^{s-1}-1)}$. Let $G_1$ and $G_2$ be the two connected components, i.e. $L_{t(2^s-1)} \setminus V' = G_1 \oplus G_2$. Then, we have

$$
\begin{aligned}
r(L^t_{t(2^s-1)}) &\leq r(G_1 \oplus G_2) + t \\
&= \max\{r(G_1), r(G_2)\} + t \quad \text{by Lemma 4.1} \\
&\leq (s-1)t + t \quad \text{by the inductive assumption} \\
&= st.
\end{aligned}
$$

$\square$

*Proof of Corollary 4.12.* Let $s$ be $\lceil \log_2(\frac{d}{t} + 1) \rceil$. By the definition of path graphs (Definition 4.2), we have $L^t_d \leq L^t_{t(2^s-1)}$ (recall that $\leq$ means being isomorphic to a subgraph). Hence, we have

$$
\begin{aligned}
r(L^t_d) &\leq r(L^t_{t(2^s-1)}) \quad \text{by Lemma 4.4} \\
&\leq s \cdot t \quad \text{by Proposition 4.11} \\
&= O(t \log d).
\end{aligned}
$$

$\square$

*Proof of Proposition 4.13.* We will prove the statement by induction on $s$.

**Base case $s = 1$:** When $s = 1$, we observe that $L^t_{t(2^1-1) \times t(2^1-1)} = L^t_{t \times t}$ is a complete graph with $t^2$ vertices. By Lemma 4.6, we have $r(L^t_{t \times t}) = t^2 \leq 4t^2 2^1$.

**Inductive step $s > 1$:** Let $d'$ be $t(2^{s-1} - 1)$. Let $V' \subset V$ be

$$
V' \triangleq \{(d' + i, j) \mid i \in [t], j \in [t(2^s - 1)]\} \cup \{(i, d' + j) \mid i \in [t(2^s - 1)], j \in [t]\}.
$$

Note that $|V'| = t^2 + 4t^2 d' \leq 4t^2 2^{s-1}$. By Lemma 4.2, we have

$$
r(L^t_{t(2^s-1) \times t(2^s-1)}) \leq r(L^t_{t(2^s-1) \times t(2^s-1)} \setminus V') + 4t^2 2^{s-1}.
$$

In other words, we first remove the vertical and horizontal "middle" stripes from the graph. By the definition of grid graphs (Definition 4.3), it is easy to check that there is no edge between $(i_1, j_1)$ and $(i_2, j_2)$ if one of the followings happens:

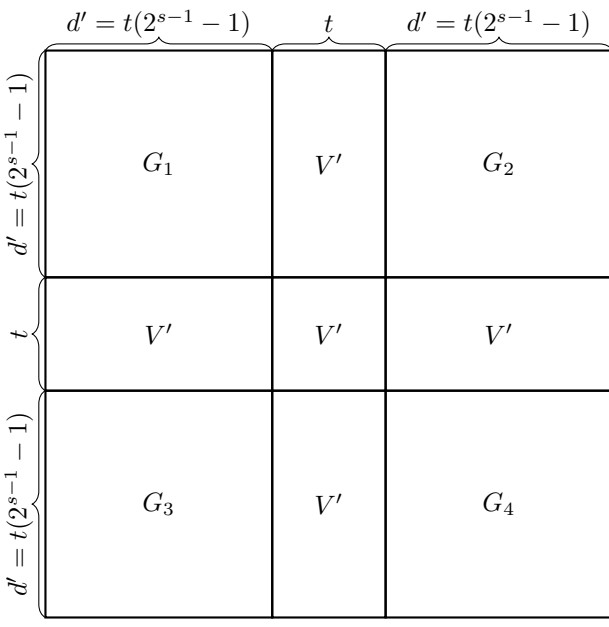

Figure 6: Illustration of $V', G_1, G_2, G_3, G_4$ in the proof of Proposition 4.13

- $i_1 = 1, \ldots, t(2^{s-1} - 1)$ and $i_2 = t(2^{s-1} - 1) + t + 1, \ldots, t(2^s - 1)$

- $j_1 = 1, \ldots, t(2^{s-1} - 1)$ and $j_2 = t(2^{s-1} - 1) + t + 1, \ldots, t(2^s - 1)$.

Therefore, there are four connected components and clearly each of them is isomorphic to $L^t_{t(2^{s-1}-1) \times t(2^{s-1}-1)}$. Let $G_1, G_2, G_3, G_4$ be the four connected components, i.e. $L^t_{t(2^s-1) \times t(2^s-1)} \setminus V' = G_1 \oplus G_2 \oplus G_3 \oplus G_4$. See figure 6 for the graphical illustration. Then, we have

$$
\begin{aligned}
r(L^t_{t(2^s-1) \times t(2^s-1)}) &\leq r(G_1 \oplus G_2 \oplus G_3 \oplus G_4) + 4t^2 2^{s-1} \\
&\leq \max\{r(G_1), r(G_2), r(G_3), r(G_4)\} + 4t^2 2^{s-1} \quad \text{by Lemma 4.1} \\
&\leq 4t^2 2^{s-1} + 4t^2 2^{s-1} \quad \text{by the inductive assumption} \\
&= 4t^2 2^s.
\end{aligned}
$$

$\square$

*Proof of Corollary 4.14.* Let $s$ be $\lceil \log_2(\frac{k}{t} + 1) \rceil$. By the definition of grid graphs (Definition 4.3), we have $L^t_{k \times k} \leq L^t_{t(2^s-1) \times t(2^s-1)}$ (recall that $\leq$ means being isomorphic to a subgraph). Hence, we have

$$
\begin{aligned}
r(L^t_{k \times k}) &\leq r(L^t_{t(2^s-1) \times t(2^s-1)}) \quad \text{by Lemma 4.4} \\
&\leq 4t^2 2^s \quad \text{by Proposition 4.13} \\
&= O(t^2 k) = O(t^2 \sqrt{d}).
\end{aligned}
$$

$\square$

*Proof of Lemma 4.15.* We will assume $V_1, \ldots, V_k$ all contain $m$ vertices and for all $i$ for all $v, v' \in V_i$, $v$ is adjacent to $v'$ (all of the blocks are fully connected). We will prove the lemma for this case, the lemma statement then follows from application of Lemma 4.4.

We will denote the vertices of $V_i$ as $v_{i,1}, \ldots, v_{i,m}$. Let $D'$ be a $t$-disintegration of $G'$. Let $D$ be a $t \times m$-tuple, where $D_{i,j} = \{v_{i,j} \mid i \in D_i, j = j\}$. If one considers $D$ in lexicographical order, then it is clearly a pre-disintegration of $G$ with length $mt$. This case then follows from Lemma C.1. $\square$

# D  Optimality of Rates with Respect to $d$ and $r$

Consider a fixed dimension $d$ and resilience $r$. Let $G$ be a graph where vertices $1, \ldots, r$ form a complete subgraph, and vertices $r + 1, \ldots, d$ are isolated (not adjacent to any other vertex). Let $L > 0$, and suppose we have an estimator that can estimate densities in $\mathcal{D}_L(G)$ at rate $O(n^{-1/(2+r-\varepsilon)})$ for some $\varepsilon > 0$. Let $p \in \mathcal{D}_{r,L}$ and $X_1, \ldots, X_r \sim p$. We will construct an estimator for $p$ using the aforementioned estimator. Define $\widetilde{X}_1, \ldots, \widetilde{X}_d \sim \widetilde{p}$ as follows:

$$(\widetilde{X}_1, \ldots, \widetilde{X}_r) \stackrel{d}{=} (X_1, \ldots, X_r),$$

$$\widetilde{X}_{r+1}, \ldots, \widetilde{X}_d \stackrel{iid}{\sim} \mathrm{Unif}([0, 1]),$$

where $(\widetilde{X}_1, \ldots, \widetilde{X}_r)$ and $(\widetilde{X}_{r+1}, \ldots, \widetilde{X}_d)$ are independent.

Note that $\widetilde{p} \in \mathcal{D}_L(G)$. Let $\hat{p}$ be an estimator for $\widetilde{p}$ such that

$$\|\widetilde{p} - \hat{p}\|_1 \in O\left(n^{-1/(r+2-\varepsilon)}\right).$$

Let $\mathcal{L}$ denote the law of a random vector. It is well known that for any function $f$ from one Euclidean space to another, $\|\mathcal{L}(f(X)) - \mathcal{L}(f(Y))\|_1 \leq \|\mathcal{L}(X) - \mathcal{L}(Y)\|_1$ (see Devroye and Lugosi (2001), Section 5.4). We will use $f : (x_1, \ldots, x_d) \mapsto (x_1, \ldots, x_r)$. Let $Y_1, \ldots, Y_d \sim \hat{p}$. Using $\mathcal{L}((Y_1, \ldots, Y_r))$ as an estimator for $p$, we have:

$$
\begin{aligned}
\|\mathcal{L}((Y_1, \ldots, Y_r)) - p\|_1 &= \|\mathcal{L}((Y_1, \ldots, Y_r)) - \mathcal{L}((X_1, \ldots, X_r))\|_1 \\
&\leq \|\mathcal{L}((Y_1, \ldots, Y_d)) - \mathcal{L}((X_1, \ldots, X_d))\|_1 \\
&= \|\hat{p} - \widetilde{p}\|_1 \\
&\in O(n^{-1/(r+2-\varepsilon)}).
\end{aligned}
$$

This shows we can estimate all densities in $\mathcal{D}_{r,L}$ at rate $O\left(n^{-1/(r+2-\varepsilon)}\right)$, while the known optimal rate is $O\left(n^{-1/(r+2)}\right)$. Thus, for any $r$ and $d$, there exists a graph with $d$ vertices and resilience $r$ such that estimation of densities in $\mathcal{D}_L(G)$ can be no faster than $O\left(n^{-1/(r+2)}\right)$.

