# OpenReview forum: "Breaking the curse of dimensionality in structured density estimation"
_NeurIPS.cc/2024/Conference — NeurIPS 2024 poster_

### Official Review · Reviewer_yapz · 2024-07-06

**Soundness:** 4
**Presentation:** 4
**Contribution:** 4
**Rating:** 6
**Confidence:** 5

**Summary:**

This paper proposed a new summary of graph, called graph resilience, which measures the complexity of density estimation. This concept is different from more its well-known counterparts, including rank, sparsity, manifold dimension etc. Quite a few concrete examples of graph along with their resilience are given. Both the concept and the theory are sound, the contribution to the literature of density estimation is solid.

**Strengths:**

The concept of resilience is novel. The definition is clear, esp Figure 3.
Section 4 is very well-written, which helps the audience a lot to understand the concept of resilience.

**Weaknesses:**

More intuitions are expected, to define $r$. The concept is beautiful, but I am curious how the authors come up with the definition. Is there is any clear intuition to define it, or the authors start from the proof of the density estimation rate and then identify this key quantity?

**Questions:**

Given an arbitrary graph, is there any concrete computational algorithm to calculate the exact resilience? If so, what’s the computational cost? I suppose the exact calculation is very expensive, that’s why the authors focused more on how to bound it.

**Limitations:**

Can the authors discuss some possible extensions? For example, what does the resilience look like for directed graphs, or for non-Markovian graphs?

---

> ### Author Rebuttal · Authors · 2024-08-06
>
> __The concept is beautiful, but I am curious how the authors come up with the definition. Is there is any clear intuition to define it, or the authors start from the proof of the density estimation rate and then identify this key quantity?__
>
> At a high level, a disintegration outlines a method to estimate a density by inductively conditioning out the entries of a random vector. For instance, consider a random vector $[X_1,X_2]$. The disintegration $(X_1)-(X_2) \Rightarrow (X_1) \Rightarrow \emptyset$ corresponds to first estimating a histogram for $X_2$, and then, for each bin $b$ of the $X_2$ histogram, estimating a histogram for $X_1|X_2 \in b$. The resilience of a graph simply characterizes the shortest disintegration, i.e. the most efficient decomposition of a distribution into factors.
>
> Removing one vertex from each component of a graph captures the idea that, after sufficient conditioning, these components become independent. This allows us to avoid estimating the high-dimensional joint density of all vertices in the graph. Instead, we can estimate the low-dimensional components individually and take their product. It's worth noting that for densities, we have the inequality:
> $\Vert\prod_{i=1}^d p_i - \prod_{j=1}^d q_j \Vert_1 \le \sum_{i=1}^d \Vert p_i - q_i \Vert_1$
>
> __Given an arbitrary graph, is there any concrete computational algorithm to calculate the exact resilience? If so, what’s the computational cost? I suppose the exact calculation is very expensive, that’s why the authors focused more on how to bound it.__
>
>
> We emphasize that it is not necessary to actually compute the exact resilience in order to benefit from it, which is one of the central aspects of this type of theory. The sample complexity is characterized by the resilience, whether or not it can be computed explicitly!
>
>
> Moreover, the reviewer is correct that in general, it is easier to bound the resilience than it is to calculate it exactly. Of course, any disintegration leads to an upper bound on r(G). The fact any nontrivial upper bound (i.e. $r(G) < d$) leads to an improvement justifies focusing on upper bounds. Of course, it is an interesting future direction to explore algorithms for its computation.
>
> __Can the authors discuss some possible extensions? For example, what does the resilience look like for directed graphs, or for non-Markovian graphs?__
>
> This is a good question! It is not clear if the same notion of graph disintegration captures the sample complexity of density estimation in other types of graphs, but we will certainly mention this as a future direction in the discussion section.

---

> > ### Comment · Reviewer_yapz · 2024-08-11
> >
> > Thanks for the response. The intuition is helpful to understand the concept. For the calculation, I understand it's not necessary to calculate it exactly for certain purpose. I asked this question because I'd like to understand the concept deeper, not for any specific (practical) purpose. Can I understand your answer as "we don't have a principled way to exactly calculate it, and we don't know the cost"?

---

> > > ### Author Response · Authors · 2024-08-13
> > > **Response to Reviewer yapz comment: On calculating graph resilience**
> > >
> > > __”Can I understand your answer as "we don't have a principled way to exactly calculate it, and we don't know the cost"?”__
> > >
> > > Correct, we don’t have a general way to calculate it exactly, and we stress that part of our contribution is to emphasize the relevance of this new quantity, so as to inspire future investigations that might provide principled algorithms. We suspect, for example, that greedy methods may work well and give useful bounds. An example of this could be greedily removing the most central vertex in each graph component according to some vertex centrality measure (e.g. vertex degree).

---

> > > > ### Comment · Reviewer_yapz · 2024-08-14
> > > >
> > > > Thanks for the clarification. I suggest the authors to clarify this computational aspect somewhere in the paper, even in the appendix. I'll keep my (positive) score.

---

### Official Review · Reviewer_HdwX · 2024-07-10

**Soundness:** 2
**Presentation:** 4
**Contribution:** 3
**Rating:** 6
**Confidence:** 3

**Summary:**

The paper studies the problem of estimating a multivariate density, assuming that the density is Markov w.r.t an underlying undirected graph $G$. It is shown that the sample complexity for estimating such a density scales with the resilience of the graph, as opposed to  the dimension d. Several examples of G are provided for which the resilience can be exactly computed or bounded.

**Strengths:**

1. The paper is written very well with a clear exposition which makes it easy to follow. The problem is well defined and motivated, along with clear notation throughout. The related work is also thorough and provides a good overview of the literature.

2. The results are to my knowledge novel, I am not aware of similar results for this problem setting. I believe the paper would be of interest for researchers in high dimensional statistics, and related problems.

**Weaknesses:**

1. It would have been helpful to provide a proof-sketch of the main theorems in the main text, this is particularly relevant here given that this is a theoretical paper. Moreover, I think it is important to outline the algorithm in the main text and not relegate it to the appendix. To handle space constraints, I think parts of Section 4 could be moved to the appendix as they are relatively less important as opposed to the aforementioned details.

2. It would have also been useful to provide some simulation results on synthetic data as this would have helped empirically validate the theoretical results.

**Questions:**

1. What is the running time of the algorithm w.r.t n, d and r? This is important to discuss in the main text and is related to the point I had raised earlier.

2. The density is assumed to be Lipschitz for the analysis, but can the results be extended to handle higher order smoothness (s times continuous differentiability)?

3. Can something be said about the optimality of the bounds? In particular, is the graph resilience the right quantity for this problem?

4. In Corollary 4.12, why not show the dependence on t explicitly in the bound?

Minor remarks:

- I think there is a typo towards the end of line 182. Another typos in line 252 ("... tend to have ...")

- In the statement of Theorem 3.1, it might be good to specify that $G$ is known.

**Limitations:**

I do not see a dicussion on limitations in the main text.

---

> ### Author Rebuttal · Authors · 2024-08-06
>
> __”It would have been helpful to provide a proof-sketch of the main theorems in the main text…”__
>
> We are happy to incorporate this into the final draft.
>
> __”It would have also been useful to provide some simulation results on synthetic data as this would have helped empirically validate the theoretical results.”__
>
> __”What is the running time of the algorithm w.r.t n, d and r?”__
>
> Please see the General Author Response.
>
> __”The density is assumed to be Lipschitz for the analysis, but can the results be extended to handle higher order smoothness (s times continuous differentiability)?”__
>
> This would involve modifying certain parts of the analysis, but we suspect that this is possible by using standard results on nonparametric estimation in function classes with higher order smoothness (say, Holder or Sobolev). We chose to stick with Lipschitz for two reasons: 1) Lipschitz is weaker than assuming higher-order smoothness, and 2) We want to keep the focus on the novel dimensional aspects of the problem through the Markov property and graph resilience.
>
> __”Can something be said about the optimality of the bounds? In particular, is the graph resilience the right quantity for this problem?"__
>
> As far as characterizing sample complexity in terms of resilience and dimension, for all dimensions $d$ and graph resilience $r$, we know that our rates cannot be improved.
>
> We _do_ know that there exists at least one class of distributions where the rate of convergence can be improved. For any Markov random fields with a tree graph, one can achieve an $L^1$ rate of convergence of $n^{-1/4}$, _asymptotically_. This corresponds to an effective dimension of $2$ and there exist trees where our results do not achieve this rate of convergence. On the other hand, our results are _uniform_ rather than asymptotic, so it is possible that our results are optimal when considering uniform deviation bounds.
>
> __”In Corollary 4.12, why not show the dependence on t explicitly in the bound?”__
>
> We can include $t$ in Corollaries 4.12 and 4.14 in the final draft. The new rates will be $O(t\log(d))$ and $O(t^2 \sqrt{d})$ respectively.

---

> > ### Comment · Reviewer_HdwX · 2024-08-12
> > **Read authors response**
> >
> > Thank you for the clarifications, I am satisfied with them. I am not insistent on the simulations, although I do think it would have strengthened the paper. But adding a proof sketch in the main text is more important given that this is a theory paper.
> >
> > I am increasing my score to 6.

---

### Official Review · Reviewer_P9wL · 2024-07-13

**Soundness:** 3
**Presentation:** 2
**Contribution:** 3
**Rating:** 5
**Confidence:** 4

**Summary:**

The authors, through the formalism of graph models for probability density functions, identify the “graph resilience” as a key quantity to estimate the sample complexity of computing the density.

**Strengths:**

The work is original and technically sound, claims are generally well supported (with one exception that I will mention later on).

The ideas of graphs resilience and disintegration are interesting and are nicely defined.

**Weaknesses:**

The only claim that I consider not well supported is related to lines 160-161, more concretely to the lower dimensional manifold. In order to support this claim, the authors should prove this claim by finding an example where the resilience r is lower than the dimension d and that cannot be mapped into a lower dimensional manifold.

The paper is not clearly written and in a way that only specialists in the field of graph modelling can understand. The relationship with classical methods like kernel density estimation is not clear.

No practical applications are shown.

From certain point of view, the novelty is not high: The authors shown that, for structured data, the Markov property is enough to simplify the complexity of density estimation. However, the structure in the data is, by itself, a correlation measure, and therefore, it is not surprising that it reduces the effect of the curse of dimensionality.

**Questions:**

Why is the manifold assumption violated in structured data if the effective dimension is lower than the embedding dimension?

Are you willing to provide a code that allows to use your findings in practical data?

**Limitations:**

No limitations are discussed. Aspects like the limited application field and the difficulties when trying to use the findings in real data should be addressed.

---

> ### Author Rebuttal · Authors · 2024-08-06
>
> __“The only claim that I consider not well supported is related to lines 160-161, more concretely to the lower dimensional manifold. In order to support this claim, the authors should prove this claim by finding an example where the resilience r is lower than the dimension d and that cannot be mapped into a lower dimensional manifold.”__
>
> __”However, the structure in the data is, by itself, a correlation measure, and therefore, it is not surprising that it reduces the effect of the curse of dimensionality.”__
>
> The way in which we are utilizing Markov random fields (MRFs) is novel and we agree that elaborating on this a bit is a good idea.
>
> A simple example of this can be found with the empty graph with $d$ vertices and no edges, which has resilience 1. In this case all of the entries of a random vector $X = (X_1,\ldots,X_d)$ must be independent and thus its density must have the form $p(x_1,\ldots, x_d) = p_1(x_1)p_2(x_2) \cdots  p_d(x_d)$. Thus the support of $p$, $\operatorname{supp}(p)$, is equal to $\operatorname{supp}(p_1) \times \cdots \times \operatorname{supp}(p_d)$ which is then a $d$-dimensional volume and hence has no low-dimensional structure whatsoever.
>
>
> As another example we might consider the $d=2$ case where $p_1$ and $p_2$ are densities that don't concentrate near a single point, e.g., they are both uniformly distributed on $[0,1]$. In order for $X_1$ and $X_2$ to lie near a 1-dimensional manifold, then $X_1$ and $X_2$ _must_ be strongly dependent and the MRF must have two vertices with an edge between them. Thus, in this case, the resilience is 2 but the manifold dimension is 1.
>
> It's worth noting that a density's graph isn't unique. The complete graph, from which one can not conclude anything regarding independence structure or a density, applies to every density. The _lack_ of edges between vertices in a Markov random field is what conveys information about a density, but removing edges tends to imply that the support of a density is filling more space.
>
> __”Are you willing to provide a code that allows to use your findings in practical data?”__
>
> __”No practical applications are shown.”__
>
> Please see the General Author Response.
>
> __”The paper is not clearly written and in a way that only specialists in the field of graph modelling can understand. The relationship with classical methods like kernel density estimation is not clear.”__
>
> Kernel density estimators, along with histograms, can achieve the minimax optimal $n^{-1/(2+d)}$ rate of convergence for Lipschitz continuous densities. An important point about kernel density estimation is that it does not adapt well to graphical structure and thus suffer from the curse of dimensionality, which is an important part of our motivation. We will include this point in our paper. Please let us know if there is some other comparison you would like to see.

---

> > ### Comment · Reviewer_P9wL · 2024-08-11
> > **Follow up**
> >
> > I thank the authors for the reply, now things are more clear.
> > Regarding practical applications, I'm still not convinced that a paper with no practical applications is worth publication in Neurips, but before deciding to maintain or raise my score I would like to have a clear reply from the authors. Is there a practical way to take advantage of resilience to compute densities? Can you show that in a real case?

---

> > > ### Author Response · Authors · 2024-08-13
> > > **Response to Reviewer P9wL comment: On practical applications**
> > >
> > > __”Is there a practical way to take advantage of resilience to compute densities? Can you show that in a real case?”__
> > >
> > > Yes, there is a practical way to take advantage of this. This is related to our explanation of graph resilience in our rebuttal to yapz. Graph resilience essentially gives the most efficient disintegration of a graph, and can be thought of as a sort of “meta-algorithm.” The disintegration gives an ordering as to how to estimate conditional densities, however it would be up to a practitioner to decide how to take care of the one-dimensional and conditional density estimation, which are themselves structured in an inductive way.
> > >
> > > More precisely, a disintegration outlines a way to inductively condition out the entries of a random vector. For instance, consider a random vector $[X_1,X_2]$. The disintegration $(X_1)-(X_2) \Rightarrow (X_1) \Rightarrow \emptyset$ corresponds to first estimating $p_{X_2}$, and then, for each $x$, estimating the density $p_{X_1|X_2=x}$. The Lipschitz condition tells us that $p_{X_1|X_2=x}$ doesn’t change drastically as $x$ changes, thus making the problem tractable.
> > >
> > > Removing one vertex from __each__ component of a graph captures the idea that, after sufficient conditioning, these components become independent. This allows us to avoid estimating the high-dimensional joint density of all vertices in the graph. Instead, we can estimate the low-dimensional components individually and take their product which is justified due to the following inequality:
> > > $\Vert\prod_{i=1}^d p_i - \prod_{j=1}^d q_j \Vert_1 \le \sum_{i=1}^d \Vert p_i - q_i \Vert_1$
> > >
> > > For a concrete example, consider the graph $(X)-(Y)-(Z)$, with the disintegration
> > >
> > >  $(X)-(Y)-(Z) \Rightarrow  (X)\quad (Z)   \Rightarrow \emptyset$.
> > >
> > > This corresponds to estimating $p_Y$, then estimating $p_{X,Z| Y}$ utilizing the fact that $X$ and $Z$ are independent given $Y$ so $p_{X,Z| Y}(x,z) = p_{X| Y}(x)p_{Z| Y}(z)$ thus instead of having to estimate  a two dimensional conditional density, one can simply estimate two one dimensional conditional densities and take the product, thereby avoiding the curse of dimensionality. This example only contains one simple instance of taking advantage of the product structure, but there may be layers of product structure, which is exemplified in our result that the graph resilience of a linear Markov graph grows only logarithmically in dimension. One also need not know the disintegration corresponding to the graph resilience in order to apply this. The resilience simply characterizes the best possible disintegration.
> > >
> > > While disintegrations are based on estimating many one-dimensional conditional distributions, one could instead estimate low-dimensional distributions. We would be happy to add an appendix (based on this discussion) outlining how a practitioner can take advantage of the disintegration structure in a more practical and readable way.

---

### Author Rebuttal · Authors · 2024-08-06

# General Author Response
We thank the reviewers for their thoughtful reviews.

Overall the response from the reviewers was quite positive:

__Presentation:__

* HdwX: "The paper is written very well with a clear exposition which makes it easy to follow."
* yaps: "The definition is clear, esp Figure 3. Section 4 is very well-written, which helps the audience a lot to understand the concept of resilience."
* HdwX & yaps: Rated presentation "4: Excellent"

__Novelty and Significance:__
* P9wl: "The ideas of graphs resilience and disintegration are interesting and are nicely defined."
* HdwX: "The results are to my knowledge novel, I am not aware of similar results for this problem setting. I believe the paper would be of interest for researchers in high dimensional statistics, and related problems."
* yapz: "This concept is different from more its well-known counterparts.... Both the concept and the theory are sound, the contribution to the literature of density estimation is solid."
* yapz: " The concept is beautiful..."

Two of the reviewers had comments regarding the algorithm in our theorems, we will address that below.

# Algorithm

__Reviewer P9wL: “Are you willing to provide a code that allows to use your findings in practical data?”__

__Reviewer HwdX: “It would have also been useful to provide some simulation results on synthetic data as this would have helped empirically validate the theoretical results.”__

__Reviewer HwdX: “What is the running time of the algorithm w.r.t n, d and r? This is important to discuss in the main text and is related to the point I had raised earlier.”__


As with most theory papers in NeurIPS, our main contribution is not a practical implementation or application to real data. Our main contribution is a novel and nontrivial analysis of the structured density estimation problem, and the introduction of a new graphical parameter that shows how the curse of dimensionality can be broken.

That being said, the algorithm in our main theorem is based on the “Scheffé tournament” estimator, which is a classical algorithm in learning theory. For example, this algorithm was also used in the recent theory paper “Nearly tight sample complexity bounds for learning mixtures of Gaussians via sample compression schemes,” which was awarded “best paper” at NeurIPS 2018 (which also did not include algorithmic details or runtime analysis). We stress that our main contribution is not this algorithm (obviously, it has been used previously), but the analysis of its performance in structured density estimation, and the surprising result that it can evade the curse of dimensionality. While the algorithm can be implemented in principle, this is not its main appeal.

Per HdwX’s suggestion we will include some discussion on the proof techniques which will help convey the computational aspect.

---

### Decision · Program_Chairs · 2024-09-25

**Decision:**

Accept (poster)

**Comment:**

This is a theory orientated paper on graphical models (i.e., high dimensional distributions whose independence structure is captured by a graph G). The paper shows that the sample complexity of learning such distributions is characterized by a novel “resilience” property of the underlying graph, which is a much richer property than previously considered properties like graph sparsity. The reviewer agreed that the paper was very well-written, and felt that the resilience property was both novel and has potential for impact on future work, especially given how well-studied graphical models are. There are ways to strengthen the paper, for example, with experiments validating the theoretical results. Nevertheless, we recommend to accept the paper on its theoretical merits alone.